# Anatomy of Massive Activations and Attention Sinks

**Shangwen Sun** [1]  **Alfredo Canziani** [1]  **Yann LeCun** [1]  **Jiachen Zhu** [1]

## Abstract

We study two recurring phenomena in Transformer language models: *massive activations*, in which a small number of tokens exhibit extreme outliers in a few channels, and *attention sinks*, in which certain tokens attract disproportionate attention mass regardless of semantic relevance. Prior work observes that these phenomena frequently co-occur and often involve the same tokens, but their functional roles and causal relationships remain unclear. Through systematic experiments, we show that the co-occurrence is largely an *architectural artifact* of modern Transformer design, and that the two phenomena serve related but distinct functions. Massive activations operate *globally*: they induce near-constant hidden representations that persist across layers, effectively functioning as implicit parameters of the model. Attention sinks operate *locally*: they modulate attention outputs across heads and bias individual heads toward short-range dependencies. We identify the pre-norm configuration as the key choice that enables the co-occurrence and show that ablating it causes the two phenomena to decouple.

## 1. Introduction

Transformer-based (Vaswani et al., 2017) Large language models (LLMs) have achieved unprecedented success across a wide range of tasks (Radford et al., 2018; 2019; Brown et al., 2020; OpenAI, 2022; Achiam et al., 2023; Touvron et al., 2023a;b; Grattafiori et al., 2024; Qwen et al., 2025; Yang et al., 2025), yet many aspects of their internal computations remain poorly understood. In this paper, we study two phenomena that reliably co-occur in decoder-only, pre-norm Transformers (Radford et al., 2018; Xiong et al., 2020): *massive activations*, in which a handful of tokens exhibit extreme outliers in a few hidden channels (Sun et al., 2024;

Yu et al., 2024a), and *attention sinks*, in which a small number of tokens attract disproportionate attention mass across many heads and layers (Xiao et al., 2024b). Both phenomena have significant practical implications for quantization (Xiao et al., 2023; Yu et al., 2024a; Son et al., 2024), pruning (Ma et al., 2023; Sandoval-Segura et al., 2026; Shin et al., 2025), KV-cache management (Ge et al., 2024; Su & Yuan, 2025; Wu & Tu, 2024), and long-context inference (Huang et al., 2023; Fu et al., 2025; Xiao et al., 2024a), among others. Understanding how these two phenomena relate is therefore both theoretically and practically important.

Prior work (Sun et al., 2024; Kaul et al., 2024; Queipo-de Llano et al., 2025) has suggested that the co-occurrence is driven by the overlap of involved tokens, but existing explanations remain largely descriptive. Here, we move beyond description to provide a *mechanistic account* of *how* and *why* this overlap emerges in pretrained LLMs. Our core finding is that the co-occurrence is not an inherent property of Transformers, but a predictable consequence of specific architectural and training choices.

We advance three central claims: **First**, normalization is a key architectural component bridging the relationship between massive activations and attention sinks. Changing the normalization configuration can suppress massive activations while preserving attention sinks. Mechanistically, massive activations interact with normalization to produce near-constant hidden representations within a forward pass, effectively serving as implicit parameters that can be exploited to generate attention sinks. **Second**, attention sinks are primarily driven by the dimensionality of the attention space and by the training context-length distribution. We further show that sinks provide a mechanism to dynamically modulate attention output across heads, biasing certain heads toward *short-range dependencies* that capture local sentence structure. **Third**, each phenomenon can be independently suppressed without degrading language-modeling performance, suggesting that their overlap reflects incidental architectural interactions rather than a functional necessity.

Together, our results clarify the causal relationship between massive activations and attention sinks and show how alternative design choices can mitigate either of the phenomena. For readability, we refer to the tokens and channels that exhibit massive activations as *spike tokens* and *spike channels*,

---

[1]New York University. Correspondence to: Shangwen Sun <shangwen.sun@nyu.edu>, Jiachen Zhu <jiachen.zhu@nyu.edu>.

*Proceedings of the 43rd International Conference on Machine Learning*, Seoul, South Korea. PMLR 306, 2026. Copyright 2026 by the author(s).

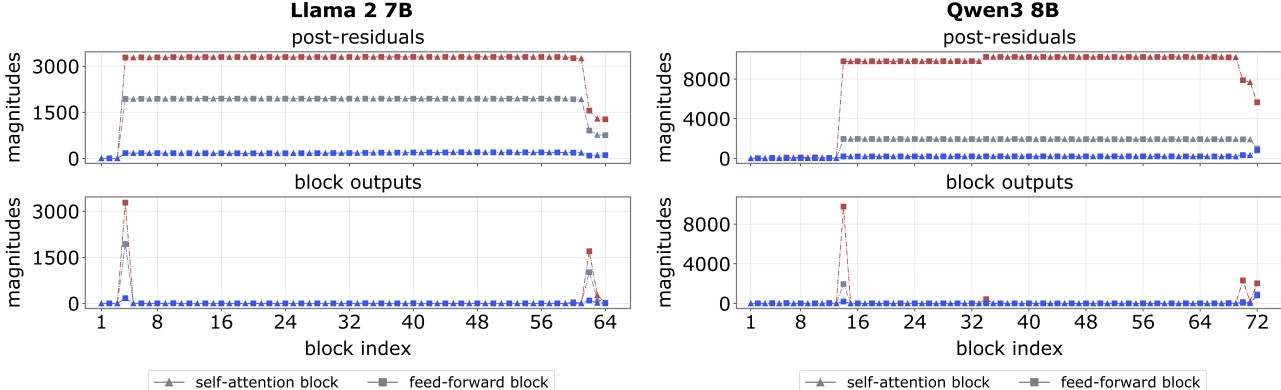

*Figure 1.* **Top-3 channel magnitudes across depth in Llama 2 7B and Qwen3 8B (post-residuals vs. block outputs).** In both models, early blocks inject massive activations that persist through most of the network before being neutralized by late blocks.

and to the tokens and attention heads affected by attention sinks as *sink tokens* and *sink heads*.

## 2. From Spikes to Sinks

This section examines the co-emergence of massive activations and attention sinks in pretrained LLMs. We begin by tracing the formation of massive activations, identifying the architectural components responsible for their generation and propagation. We then show how normalization transforms these tokens with massive activations into sparse, near-constant input vectors, enabling the formation of attention sinks. Our analysis draws primarily from the Llama (Touvron et al., 2023a;b; Grattafiori et al., 2024) and Qwen (Qwen et al., 2025; Yang et al., 2025) model families. We present the main findings and key empirical results here, deferring additional supporting evidence to Appendix D. The mathematical notation and architectural preliminaries required are summarized in Appendix A.

Some findings discussed in this section and the next are restatements or extensions of prior work. We provide a detailed account of their relation to prior work in Section 4.

### 2.1. The Emergence of Massive Activations

Prior work (Bondarenko et al., 2023; Xiao et al., 2023; Nrusimha et al., 2024; Sun et al., 2024; Yu et al., 2024a) has characterized massive activations as extreme outliers in the post-residual hidden representations of Transformers. These outliers, which often exceed typical activation scales by several orders of magnitude, exhibit five recurring properties across models and prompts:

  (i) They appear only in *intermediate layers*.
 (ii) They appear only in a *small number of channels*.
(iii) The affected channels *consistently spike together*.
 (iv) The spikes maintain *almost fixed inter-channel ratios*.
  (v) They appear only in a *small number of tokens*.

In this subsection, we trace the emergence of massive activations systematically, showing how each of these properties arises. As we will see in the next subsection, these properties play a foundational role in enabling attention sinks.

### 2.1.1. THE LIFE CYCLE OF MASSIVE ACTIVATIONS

To characterize how massive activations vary with depth, we track the top-3 channel magnitudes of post-residual hidden representations (Figure 1, top panels), following Sun et al. (2024). The magnitudes follow a "rise–plateau–fall" trajectory: a sharp increase in early blocks, a long plateau through intermediate blocks, and an abrupt return to typical magnitudes near the end. This suggests a three-stage life cycle: (1) early blocks inject extreme values into the hidden representations; (2) intermediate blocks propagate these values via the residual connection; and (3) late blocks neutralize them by inject extreme values with opposite sign. We describe each stage in turn.

**Step-up blocks.** By examining the individual block outputs (Figure 1, bottom panels), we find that massive activations are reliably introduced by *one or two early blocks*, which we term *step-up blocks*. Prior to these blocks, spike tokens have magnitudes comparable to standard tokens. The step-up blocks produce extreme values in the spike channels, which are then added to the hidden representation via the residual connection, creating the massive activations.

**Residual accumulation.** In pre-norm Transformers, the hidden representation at depth $i$ can be expressed by unrolling the recurrence in Equation (17):

$$\mathbf{H}_{i+1} = \mathbf{H}_1 + \sum_{j=1}^{i} \mathcal{F}_j(\mathrm{RMSNorm}(\mathbf{H}_j)). \qquad (1)$$

Because the residual stream is additive, extreme values injected by any block $\mathcal{F}_j$ persist through all subsequent blocks unless explicitly counteracted. Empirically, intermediate block contributions to spike channels are typically two to three orders of magnitude smaller than the massive activations themselves. The massive activations dominate the residual stream until a later block cancels them.

**Step-down blocks.** As shown in Figure 1, massive activations consistently disappear near the end of the network. We identify *one or a few late blocks*, termed *step-down blocks*,

*Table 1.* **Step-up and step-down block indices across models.** Step-up blocks appear near the beginning and step-down blocks near the end, confining massive activations to intermediate layers.

| MODEL | # BLOCKS | STEP-UP | STEP-DOWN |
|---|---|---|---|
| LLAMA 2 7B | 64 | 4 | 62 |
| LLAMA 2 13B | 80 | 8 | 78, 79 |
| LLAMA 3 8B | 64 | 4 | 64 |
| QWEN2.5 7B | 56 | 8, 10 | 54, 55 |
| QWEN2.5 14B | 96 | 10 | 90, 92, 94, 95 |
| QWEN3 8B | 72 | 14 | 70, 72 |
| QWEN3 14B | 80 | 14 | 79 |

whose outputs match the massive activations in magnitude but carry the opposite sign in the corresponding channels. These blocks neutralize the massive activations, returning the hidden representation to a standard range.

Section 2.1.1 summarizes the step-up and step-down block indices across models. The consistent positioning of step-up blocks near the beginning and step-down blocks near the end directly accounts for **Property (i)**: massive activations are confined to intermediate layers because they are injected early and systematically neutralized before the final output.

### 2.1.2. FEED-FORWARD BLOCK AS DIRECTIONAL QUADRATIC AMPLIFIER

While both attention and feed-forward blocks possess the theoretical capacity to produce large outputs, our analysis reveals that the SwiGLU-based feed-forward block is the primary source of massive activations, functioning as a *directional quadratic amplifier*. We characterize the mechanism by which extreme activations arise for a small subset of tokens in Llama 2 7B. Other models share the same high-level mechanism, but differ in more precise details that instantiate it. we defer those results to Appendix D.

**Near-identity gating regime.** Let $\tilde{\mathbf{h}}^{(s)} \in \mathbb{R}^{d_{\text{model}}}$ denote the normalized input of a spike token to a step-up or step-down feed-forward block. We empirically observe that the SiLU nonlinearity operates in a near-identity regime ($\text{SiLU}(\mathbf{x}) \approx \mathbf{x}$), as shown in Figure 2. Under this approximation, the feed-forward transformation reduces to:

$$\mathcal{F}_{\text{ffn}}(\tilde{\mathbf{h}}^{(s)}) \approx \mathbf{W}_{\text{down}} \cdot \Big( (\mathbf{W}_{\text{gate}} \tilde{\mathbf{h}}^{(s)}) \odot (\mathbf{W}_{\text{up}} \tilde{\mathbf{h}}^{(s)}) \Big). \quad (2)$$

**High-gain quadratic structure.** Let $\mathbf{W}_{\text{gate}}^{(i)}$ and $\mathbf{W}_{\text{up}}^{(i)}$ denote the $i$-th rows of the respective weight matrices, and let $\mathbf{W}_{\text{down}}^{(k,i)}$ denote the $(k,i)$-th entry of $\mathbf{W}_{\text{down}}$. Each output coordinate $k$ then admits the quadratic form (derived in detail in Theorem C.2):

$$\mathcal{F}_{\text{ffn}}(\tilde{\mathbf{h}}^{(s)})_k \approx \tilde{\mathbf{h}}^{(s)\top} \mathbf{U}_k \tilde{\mathbf{h}}^{(s)} = \tilde{\mathbf{h}}^{(s)\top} \mathbf{S}_k \tilde{\mathbf{h}}^{(s)}, \quad (3)$$

where

$$\mathbf{U}_k = \sum_{i=1}^{d_{\text{ffn}}} \mathbf{W}_{\text{down}}^{(k,i)} \, \mathbf{W}_{\text{gate}}^{(i)} \, \mathbf{W}_{\text{up}}^{(i)\top}, \quad (4)$$

$$\mathbf{S}_k = \tfrac{1}{2}(\mathbf{U}_k + \mathbf{U}_k^\top). \quad (5)$$

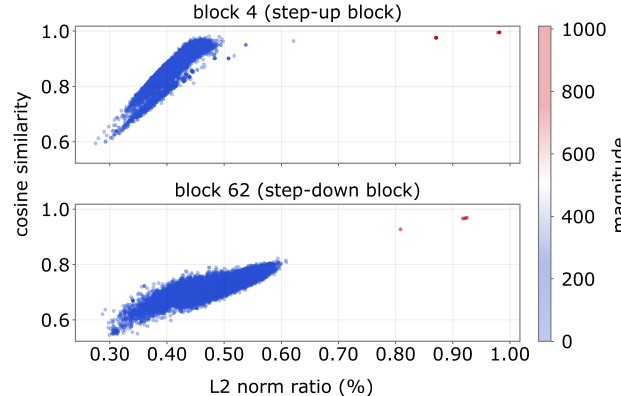

*Figure 2.* **Input-output characteristics of** SiLU **in step-up and step-down blocks of Llama 2 7B.** Based on 1024 randomly sampled sentences from C4 dataset (Raffel et al., 2020), we plot the cosine similarity and norm ratio for each token. Points are colored by the maximum magnitude of the block output. For spike tokens (red points), both direction and norm remain largely unchanged.

Figure 3 shows the Frobenius norms $\|\mathbf{U}_k\|_F$ across all output coordinates and feed-forward blocks of Llama 2 7B. Spike channels correspond precisely to coordinates with exceptionally large $\|\mathbf{U}_k\|_F$, and these high-norm coordinates appear exclusively in step-up and step-down blocks. Inspection of the weight matrices reveals that, for high-gain channels $k$, $\mathbf{W}_{\text{down}}$ contains anomalously large entries $\mathbf{W}_{\text{down}}^{(k,i)}$ for certain intermediate dimensions $i$, and the corresponding rows $\mathbf{W}_{\text{gate}}^{(i)}$ and $\mathbf{W}_{\text{up}}^{(i)}$ are highly collinear, consistent with previous observations in (Yu et al., 2024a; Yang et al., 2024a; Fishman et al., 2025).

**Rank-one dominance.** Figure 4 compares the eigenvalue spectra of $\mathbf{S}_k$ for spike versus non-spike channels. For spike channels, $\mathbf{S}_k$ is dominated by a single eigenvalue $\lambda_\star$ whose magnitude far exceeds the rest of the spectrum. Let $\mathbf{s}_\star$ denote the corresponding unit eigenvector. In such cases, the feed-forward block then acts as a *directional quadratic*

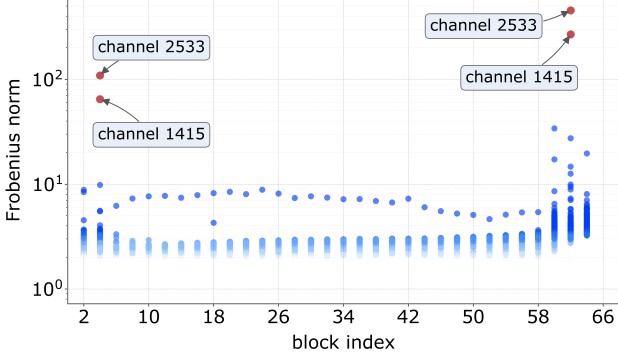

*Figure 3.* **Frobenius norms** $\|\mathbf{U}_k\|_F$ **for the quadratic forms in Llama 2 7B.** Spike channels align with $\mathbf{U}_k$ matrices that have substantially larger norms than typical channels. These high-norm coordinates appear exclusively in step-up and step-down blocks.

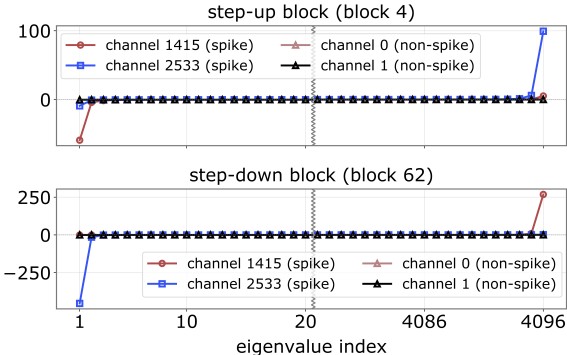

**Figure 4.** **Eigenvalue spectra of $\mathbf{S}_k$ for spike vs. non-spike channels in Llama 2 7B.** Spike channels exhibit a single dominant eigenvalue $\lambda_\star$ that is orders of magnitude larger than the remainder of the spectrum; non-spike channels show no such outlier.

*amplifier* for these channels:

$$\mathcal{F}_{\text{ffn}}(\tilde{\mathbf{h}}^{(s)})_k \approx \tilde{\mathbf{h}}^{(s)\top}\mathbf{S}_k\tilde{\mathbf{h}}^{(s)} \tag{6}$$

$$\approx \lambda_\star(\mathbf{s}_\star^\top\tilde{\mathbf{h}}^{(s)})^2 \tag{7}$$

$$= \lambda_\star\sqrt{\text{d}_{\text{model}}}\cos(\mathbf{s}_\star, \tilde{\mathbf{h}}^{(s)}). \tag{8}$$

When the input $\tilde{\mathbf{h}}^{(s)}$ aligns with the spike direction $\mathbf{s}_\star$, the squared projection is amplified by $\lambda_\star$, producing massive activations. Crucially, inspection of the spike directions across all spike channels reveals that their $\mathbf{S}_k$ matrices share nearly the same principal eigenvector $\mathbf{s}_\star$. Consequently, when an input aligns with this common spike direction, all spike channels are activated simultaneously.

This analysis accounts for **Property (ii)**: the scarcity of high-gain quadratic forms explains why massive activations are confined to a small subset of channels. Furthermore, the existence of a shared spike direction across these channels underpins **Properties (iii), (iv) and (v)**; specifically, it accounts for the synchronized triggering of affected channels and their invariant activation magnitude ratios, which are governed by the leading eigenvalues of $\mathbf{S}_k$. Finally, because these spike directions are restricted to a highly localized region of the high-dimensional space $\mathbb{R}^{\text{d}_{\text{model}}}$, extreme activations occur only for tokens whose representations closely align with $\mathbf{s}_\star$. For the vast majority of tokens, the projection onto this direction is negligible.

### 2.1.3. WHAT MAKES A TOKEN A SPIKE TOKEN

While the feed-forward block provides the *capacity* for amplification, it requires the $\tilde{\mathbf{h}}^{(s)}$ to align with the trigger direction $\mathbf{s}_\star$ in order to generate massive activations. Prior work (Sun et al., 2024) establishes that spike tokens are almost exclusively the first tokens or delimiter tokens; we now study why these tokens consistently achieve such alignments.

**First tokens.** The initial position serves as the most consistent catalyst for massive activations. A vocabulary-wide probe (Table 2) reveals that over $98\%$ of vocabulary items manifest as spike tokens when placed at position $0$, but rarely do so at subsequent indices. This disparity confirms that

*Table 2.* **Ubiquity of initial spikes across diverse LLMs.** For nearly all evaluated models, positional occupancy at the initial position induces massive activations in intermediate layers, independent of the token's semantic identity.

| MODEL | # VOCAB | # SPIKE TOKEN | RATIO |
|---|---|---|---|
| LLAMA 2 7B | 32,000 | 31,887 | 99.65% |
| LLAMA 2 13B | 32,000 | 31,889 | 99.65% |
| LLAMA 3 8B | 128,256 | 127,956 | 99.77% |
| QWEN 2.5 7B | 152,064 | 149,587 | 98.40% |
| QWEN 2.5 14B | 152,064 | 149,645 | 98.40% |
| QWEN 3 8B | 151,936 | 151,830 | 99.93% |
| QWEN 3 14B | 151,936 | 151,824 | 99.93% |

the phenomenon is driven by architectural position rather than token semantics. The few exceptions are primarily rare characters from low-resource scripts; we find that their embeddings close to the initialization values, likely due to infrequent gradient updates during pre-training.

The behavior of the initial position arises because the attention block collapses to a simple linear map. Since the first token only attend to itself, its output reduces to:

$$\mathcal{F}_{\text{attn}}(\mathbf{h}^{(1)}) = \sum_{i=1}^{\text{N}_{\text{head}}}\mathbf{W}_O^{(i)\top}\mathbf{W}_V^{(i)\top}\mathbf{h}^{(1)} \equiv \mathbf{W}_{\text{VO}}^\top\mathbf{h}^{(1)}, \tag{9}$$

where $\mathbf{h}^{(1)} \in \mathbb{R}^d$ is the hidden state of the first token, and $\mathbf{W}_{\text{VO}} \coloneqq \sum_{i=1}^{\text{N}_{\text{head}}}\mathbf{W}_V^{(i)}\mathbf{W}_O^{(i)}$ is the linear mapping matrix.

In this regime, the attention block applies a *static linear transformation* that is identical across all prompts, consistently steering the first tokens' representations toward the trigger direction $\mathbf{s}_\star$ and thereby inducing the massive activations observed in intermediate layers.

**Delimiter Tokens.** Tokens such as periods and newlines follow a mechanistic trajectory similar to first-token sinks. In the early attention blocks, these tokens exhibit significantly elevated post-RMSNorm magnitudes, stemming from the near-collinearity of their embeddings with the learned scaling parameters of RMSNorm. This magnitude surge induces attention heads to allocate disproportionate weight to the token itself, regardless of the preceding context. So delimiter tokens emulate the isolated environment of the first token across multiple heads. This *self-sinking* behavior allows static linear transformations to project their latent states toward the same high-gain manifold as the first token. Once aligned with $\mathbf{s}_\star$, these representations undergo directional quadratic amplification.

In summary, a token transitions into a spike token when it demonstrates a strong self-sinking bias in early layers, establishing the stable linear trajectory required to activate the directional quadratic amplifier.

### 2.2. The Emergence of Attention Sinks

Having traced the generation and propagation of massive activations, we now characterize how these spike tokens induce the *attention sink* phenomenon. Specifically, we



*Figure 5.* **Cosine similarity among spike tokens before and after step-up block in Llama 2 7B.** Pre-step-up representations vary across spike tokens, but post-step-up representations collapse to nearly identical directions, validating the constant approximation.

demonstrate that normalization transforms spike tokens into sparse, bounded, and nearly constant input vectors, enabling the formation of attention sinks.

### 2.2.1. NORMALIZATION TRANSFORMS SPIKE TOKENS

In pre-norm Transformer architectures, each attention block operates on normalized hidden representations. Let $\mathbf{h}^{(s)}$ denote the hidden representation of a spike token and let $\tilde{\mathbf{h}}^{(s)}$ denote the output of $\mathrm{RMSNorm}\left(\mathbf{h}^{(s)}\right)$. The transformation imparts three properties central to attention sink formation.

**Bounded Range.** Normalization suppresses the extreme magnitudes of spikes, mapping the representation to a bounded range (proof deferred to Theorem C.3):

$$|\tilde{\mathbf{h}}_i^{(s)}| \leq \sqrt{\mathrm{d}_{\mathrm{model}}}, \quad \forall i \in \{1, \ldots, \mathrm{d}_{\mathrm{model}}\}. \quad (10)$$

Hence, even if the pre-norm input contains values on the magnitudes of thousands, the block output $\tilde{\mathbf{h}}^{(s)}$ remains moderate and numerically stable.

**Sparsification.** Because the norm $\|\mathbf{h}^{(s)}\|$ is dominated by a few outlier coordinates, the normalization process effectively suppresses non-spike channels. Consequently, the normalized state $\tilde{\mathbf{h}}^{(s)}$ can be approximated as:

$$\tilde{\mathbf{h}}^{(s)} \approx \sum_{i \in \mathcal{C}} \tilde{\mathbf{h}}_i^{(s)} \mathbf{e}_i, \quad (11)$$

where $\mathcal{C}$ denotes the set of spike channel indices and $\mathbf{e}_i$ represents the $i$-th standard basis vector. This transformation yields a sparse, approximately multi-hot representation that is concentrated within a low-dimensional subspace of the original embedding space.

**Near-constant vector.** Spike channels maintain nearly fixed magnitude ratios across spike tokens (**Property (iv)**), so the normalized values $\tilde{\mathbf{h}}_i^{(s)}$ for $i \in \mathcal{C}$ are approximately token-invariant. Consequently, for any spike tokens $a$ and $b$:

$$\mathrm{RMSNorm}(\mathbf{h}^{(a)}) \approx \mathrm{RMSNorm}(\mathbf{h}^{(b)}), \quad (12)$$

even when $\mathbf{h}^{(a)}$ and $\mathbf{h}^{(b)}$ differ substantially in their non-spike channels. Normalization thus collapses distinct representations into a *near-constant* sparse vector, largely erasing token-specific variation. This collapse is empirically demonstrate in Figure 5, where spike tokens following the step-up blocks exhibit cosine similarities approaching 1.0.

### 2.2.2. GEOMETRIC ALIGNMENT CREATES SINKS

Spike tokens produce sparse normalized representations, which severely restrict the dimensionality of their resulting attention projections. For a given head, the key vector $\mathbf{k}^{(s)}$ of a sink token is given by:

$$\mathbf{k}^{(s)} = \mathbf{W}_K^\top \tilde{\mathbf{h}}^{(s)} \approx \sum_{i \in \mathcal{C}} \tilde{\mathbf{h}}_i^{(s)} \mathbf{W}_K^\top \mathbf{e}_i, \quad (13)$$

where $\mathbf{W}_K^\top \mathbf{e}_i$ corresponds to the $i$-th row of the weight matrix. Consequently, the keys $\mathbf{k}^{(s)}$ are confined to the span of only a few rows. In practice, we find this subspace typically collapses to only one or two dimensions—a significant reduction compared to the full head dimension $\mathrm{d}_{\mathrm{head}}$.

Although empirical analysis shows that non-sink queries $\mathbf{q}^{(n)}$ and keys $\mathbf{k}^{(n)}$ also reside in a constrained subspace, their manifold is significantly more expansive than that of the spike tokens. We posit that the emergence of an attention sink is determined not by the absolute volume of these subspaces, but by their *relative geometric alignment*:

- **Sink Heads:** The $\mathbf{q}^{(n)}$ subspace is positioned closer to the fixed $\mathbf{k}^{(s)}$ than to the $\mathbf{k}^{(n)}$ subspace. This alignment produces large, consistent logit gaps in favor of the sink token across diverse inputs.
- **Non-Sink Heads:** The $\mathbf{q}^{(n)}$ subspace is more closely aligned with its non-sink keys $\mathbf{k}^{(n)}$, resulting in attention patterns that distribute mass according to token semantics rather than a fixed default position.

As visualized via t-SNE (Maaten & Hinton, 2008) in Figure 6, the difference between sink and non-sink heads lies in this subspace alignment. In sink heads, the model exploits the near-constant nature of spike keys to create a stable default position for attention mass, effectively offloading excess attention weight to a token whose representation has been neutralized by the normalization function.

Attention sinks arise from two properties of spike tokens after normalization: sparsity and near-constancy. Sparsity restricts sink keys to a low-dimensional subspace (often one or two dimensions) of the row space of $\mathbf{W}_K$. Near-constancy keeps those keys nearly invariant across prompts. Together, these properties allow the model to reliably separate sink keys from non-sink keys into distinct subspaces, and this separation manifests as the logit gaps characteristic of attention sinks.

### 2.3. Summary of Findings

This section links massive activations and attention sinks through an architecture-driven pathway in pre-norm Transformers. Massive activations originate from a small number of early *step-up* feed-forward blocks. In these blocks, SwiGLU behaves as a *directional quadratic amplifier*: rare high-gain quadratic forms share a common trigger direction, and when a token aligns with it, the token becomes a token

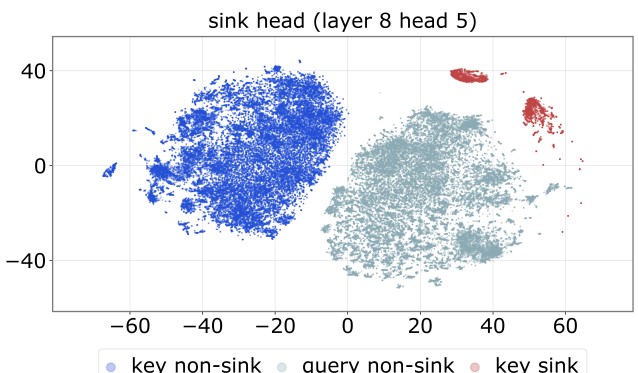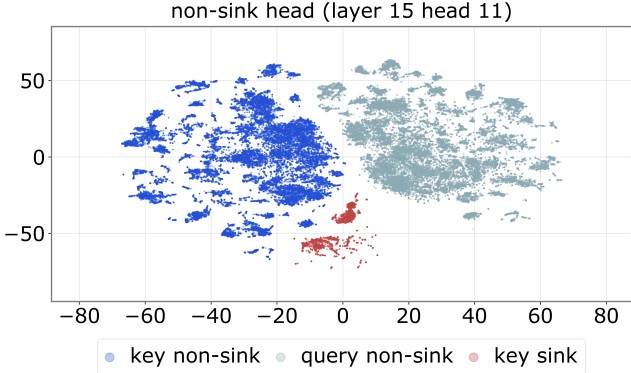

*Figure 6.* **t-SNE visualization of query and key vectors for a representative sink head (left) and non-sink head (right).** In the sink head, sink keys $\mathbf{k}^{(s)}$ lie closer to $\mathbf{q}^{(n)}$ than non-sink keys $\mathbf{k}^{(n)}$, creating large logit gaps. In the non-sink head, $\mathbf{k}^{(s)}$ and $\mathbf{k}^{(n)}$ are approximately equidistant from $\mathbf{q}^{(n)}$, preventing the formation of a privileged sink position.

carrying massive activations. Because the residual stream is additive, these outliers persist across intermediate layers.

Normalization then maps spike-token representations to inputs that are sparse and nearly constant. As a result, diverse spike tokens collapse to a same vector, making their keys low-dimensional and nearly invariant across prompts. The learned key projection $\mathbf{W}_K$ consequently maps spike keys and non-spike keys into distinct subspaces. Attention sinks then emerge in heads whose query subspace aligns more strongly with the fixed sink-key subspace than with the non-sink-key subspace, creating the persistent logit gaps that define attention sinks. This completes the account of how massive activations and attention sinks co-emerge.

## 3. Anatomy of Spikes and Sinks

The previous section characterizes how massive activations and attention sinks co-emerge in pretrained LLMs, suggesting that both phenomena arise from the interactions between architectural components and learned weights. We now shift from mechanism to causality. Guided by our findings, we perform targeted ablations to identify which architectural and training choices modulate these phenomena, ultimately establishing the causal relationship between the two.

**Experimental setup.** For our baseline setup, we train a Llama-style 7B model (Touvron et al., 2023a;b; Grattafiori et al., 2024) from scratch on the DCLM dataset (Li et al., 2024) with a budget of 100B tokens. We follow the standard Llama training recipe, supplementing unspecified details with established open-source implementations. Despite training on substantially fewer tokens than the original Llama models, we successfully reproduce the massive activations and attention sinks phenomena described previously. In each ablation, we modify a specific set of settings while keeping the remaining configuration fixed. We evaluate the models using sentences randomly sampled from the C4 dataset (Raffel et al., 2020) and report perplexity, sink ratio (Gu et al., 2025), and maximal activation magnitudes. Training and evaluation details are provided in Appendix B.

### 3.1. Ablating Optimization Hyperparameters

Before proceeding to targeted architectural ablations, we examine the sensitivity of these phenomena to common training hyperparameters: learning rate, weight decay, AdamW (Loshchilov & Hutter, 2017) momentum ($\beta_2$), and total training tokens. Results are summarized in Table 3.

Two patterns emerge. First, while the sink ratio is not strictly monotonic in perplexity, it remains a robust proxy for *optimization health*. Suboptimal configurations—such as extreme learning rates, disabling weight decay, or misspecified $\beta_2$—consistently reduce the sink ratio. Conversely, favorable configurations—such as extending training bud-

*Table 3.* **Optimization Hyperparameters ablations.** We observe that the sink ratio could serve as a proxy for optimization health. Conversely, the magnitude of massive activations varies largely independently of both perplexity and the sink ratio. Highlighted rows denote the baseline result.

| SETUP | PERPLEXITY | SINK RATIO | SPIKE |
|---|---|---|---|
| BASE LEARNING RATE | | | |
| $7.5 \times 10^{-5}$ | 11.8 | 18.6% | 1,447 |
| $1.5 \times 10^{-4}$ | 10.7 | 31.8% | 2,251 |
| $3.0 \times 10^{-4}$ | 10.1 | 46.0% | 3,818 |
| $6.0 \times 10^{-4}$ | 10.0 | 51.5% | 3,773 |
| $1.2 \times 10^{-3}$ | 10.2 | 39.2% | 2,723 |
| MINIMAL LEARNING RATE | | | |
| $3 \times 10^{-5}$ | 10.1 | 46.0% | 3,818 |
| $3 \times 10^{-4}$ | 10.7 | 56.8% | 2,870 |
| WEIGHT DECAY | | | |
| 0.0 | 10.4 | 33.8% | 12,275 |
| 0.1 | 10.1 | 46.0% | 3,818 |
| ADAMW $\beta_2$ | | | |
| 0.9 | 10.1 | 49.0% | 2,832 |
| 0.95 | 10.1 | 46.0% | 3,818 |
| 0.999 | 10.7 | 20.9% | 1,855 |
| TRAINING TOKENS | | | |
| 100B | 10.1 | 46.0% | 3,818 |
| 200B | 9.5 | 63.3% | 1,848 |

*Table 4.* **Feed-forward block design ablations.** Massive activations and attention sinks emerge across all evaluated designs, including attention-only and single linear configurations. Notably, GeLU and SwiGLU architectures yield significantly higher spike magnitudes by acting as efficient amplifiers. In contrast, linear and attention-only blocks exhibit much lower spikes. Highlighted rows denote the baseline result.

| Setup | Perplexity | Sink Ratio | Spike |
|---|---|---|---|
| Feed-Forward Block | | | |
| GeLU | 10.1 | 69.3% | 3,369 |
| Linear | 12.5 | 58.9% | 688 |
| Attention | 10.8 | 73.9% | 637 |
| SwiGLU | 10.1 | 46.0% | 3,818 |

*Table 5.* **Normalization configuration ablations.** Applying post-block normalization (Sandwich) or element-wise transformations (DynamicTanh) successfully suppresses massive activations. Notably, the model still maintains a significant sink ratio through alternative strategies, demonstrating that sinks can exist independently of massive activations. Highlighted rows denote the baseline result.

| Setup | Perplexity | Sink Ratio | Spike |
|---|---|---|---|
| Normalization | | | |
| Sandwich | 9.8 | 44.7% | 520 |
| Sandwich (QK) | 10.0 | 42.0% | 92 |
| DynamicTanh | 10.0 | 61.0% | 153 |
| Pre-Norm | 10.1 | 46.0% | 3,818 |

get or disabling learning-rate decay—substantially increase the sink ratio. This suggests that the intensity of attention sinks is tied to the overall optimization health. Second, the magnitudes of massive activations vary independently of perplexity and sink ratio. For instance, disabling weight decay causes activation spikes to exceed 12,000 without any corresponding change in sink ratio or perplexity. spikes drive normalized representations into a sparse, near-constant regime, after which further growth in magnitudes contribute diminishingly to attention sinks. Having established that sinks and spikes respond differently to optimization configurations, we now consider architectural interventions that directly target their underlying mechanisms.

## 3.2. Ablating Massive Activations

In the previous section, we identify two architectural components that strongly affect the emergence of massive activations: (1) the SwiGLU-based feed-forward network, which generates the massive activations, and (2) the normalization configuration, which governs their propagation and maps spike tokens to sparse, near-constant vectors. In this subsection, we ablate both design choices.

### 3.2.1. Feed-Forward Block Design

Our earlier analysis traced the origin of massive activations to the SwiGLU blocks. To test whether this specific design is a prerequisite for both phenomena, we ablate the feed-forward architecture. Specifically, we evaluate the standard two-layer GeLU-based feed-forward block used in the original Transformer (Vaswani et al., 2017), a simplified single linear layer, and an attention-only configuration where all feed-forward blocks are replaced with additional attention layers. Results are summarized in Table 4. The results indicate that massive activations and attention sinks emerge across all configurations, suggesting that the specific feed-forward block design is not the primary causal driver of either phenomenon. The specific block design is therefore not a prerequisite, but it is a strong modulator of amplification efficiency. SwiGLU and GeLU concentrate outlier growth within a single step, while linear and attention blocks require gradual accumulation across layers.

### 3.2.2. Normalization Configuration

Normalization shapes these phenomena along two axes: how outliers accumulate in the residual stream (governed by the pre-norm configuration), and how spike tokens are transformed into sparse, near-constant vectors (governed by the normalization operator itself). We probe both axes through three variants. First, we test sandwich normalization (Ding et al., 2021) and a variant utilizing QK-Norm (Olmo et al., 2025). Second, we replace standard normalization with an element-wise transformation, DynamicTanh (Zhu et al., 2025; Chen et al., 2025), which lacks the mathematical capacity to map extreme outliers into sparse, near-constant vectors. Table 5 summarizes these results.

The results demonstrate that normalization directly decouples sinks from spikes. Sandwich normalization reduces spikes while preserving sink ratio nearly identical to baseline. Because the extra RMSNorm layer bounds the block output, it prevents the residual stream from accumulating the unbounded values necessary for massive outliers. Replacing the block-level norm with QK-Norm almost entirely eliminates spikes, confirming that these outliers are primarily generated to influence the query and key projections.

Conversely, as observed by Owen et al. (2025a), element-wise transformations like DynamicTanh also prevent the emergence of massive activations entirely. This aligns with our hypothesis: because DynamicTanh is bounded and operates element-wise rather than via a vector-wide norm, it cannot facilitate the creation of sparse vectors from high-magnitude spikes. Interestingly, the DynamicTanh model yields the highest sink ratio while maintaining low spike magnitudes. These results confirm that while massive spikes are an artifact of specific normalization configurations, they are not a prerequisite for attention sinks.

## 3.3. Ablating Attention Sinks

Building on our earlier analysis, we find that sink formation depends critically on whether sink and non-sink keys can occupy geometrically separable subspaces. We therefore begin by ablating per-head representational capacity, which determines whether the attention subspace has sufficient room to segregate sink keys from non-sink keys. We then conduct

*Table 6.* **Attention head settings ablations.** Head dimension is the primary architectural driver of sink formation; larger dimensions provide the capacity for the attention subspace to isolate the sink keys. Concentrating capacity into fewer, larger heads intensifies sink behavior. Highlighted rows denote the baseline configuration.

| SETUP | PERPLEXITY | SINK RATIO | SPIKE |
|---|---|---|---|
| NUMBER OF HEADS | | | |
| 8 | 10.4 | 37.1% | 1,253 |
| 16 | 10.3 | 41.7% | 1,936 |
| 32 | 10.1 | 46.0% | 3,818 |
| HEAD DIMENSION | | | |
| 8 | 11.3 | 4.1% | 291 |
| 16 | 10.8 | 9.8% | 315 |
| 32 | 10.5 | 27.9% | 829 |
| 64 | 10.3 | 37.7% | 2,112 |
| 128 | 10.1 | 46.0% | 3,818 |
| HEAD DIM / NUMBER OF HEADS | | | |
| 8/512 | 10.7 | 11.0% | 1,205 |
| 16/256 | 10.4 | 30.8% | 1,750 |
| 32/128 | 10.3 | 41.1% | 1,916 |
| 64/64 | 10.2 | 44.1% | 2,523 |
| 128/32 | 10.1 | 46.0% | 3,818 |
| 256/16 | 10.1 | 52.1% | 3,429 |

two further ablations motivated by prior work: one on gated attention (Qiu et al., 2025), which has been shown to reduce attention sinks and massive activations, and one on context length, motivated by (Xiao et al., 2024a), which argues that attention sinks primarily bias short-range dependence. We ablate all three factors in turn below.

### 3.3.1. ATTENTION HEAD SETTINGS

Our earlier findings identified the segregation of sink and non-sink keys as the primary driver of sink formation. Since this mechanism is inherently tied to per-head capacity, we systematically ablate total head count, head dimension, and head factorization to disentangle their individual contributions. Results are summarized in Table 6.

The results confirm that *head dimension* is the dominant architectural factor governing sink emergence. Increasing $d_{head}$ from 8 to 128 produces a monotonic rise in both the sink ratio and spike magnitude, supporting our geometric hypothesis: larger head dimensions expand the attention subspace sufficiently to cleanly separate sink keys from non-sink keys, enabling the generation of a large logit gap.

### 3.3.2. GATED ATTENTION

Following (Qiu et al., 2025), we employ gated attention variants to test the hypothesis that dynamic multiplicative routing can destabilize or prevent attention sink formation. As shown in Table 7, gating conditioned on the current hidden representation drastically suppresses the sink ratio and effectively eliminates spikes, with minimal impact on perplexity. By contrast, unconditional gating or gating tied to static signals (such as position or token embedding) preserves

*Table 7.* **Gated attention ablations.** Conditional gating—where the gate is a function of current representation—eliminates the need for attention sinks when applied per channel or per head. This suggests that sinks function as a "learned gate" to balance head contributions. Unconditional or static gates (positional or token-based) fail to suppress sinks, as they lack the dynamic, input-dependent routing necessary to substitute for sink behavior.

| SETUP | PERPLEXITY | SINK RATIO | SPIKE |
|---|---|---|---|
| CONDITIONAL GATING | | | |
| CHANNEL | 10.0 | 4.5% | 202 |
| HEAD | 10.1 | 6.4% | 186 |
| SINGLE | 10.2 | 31.2% | 316 |
| UNCONDITIONAL GATING | | | |
| CHANNEL | 10.1 | 42.2% | 1,922 |
| HEAD | 10.1 | 41.3% | 1,884 |
| SINGLE | 10.2 | 44.3% | 1,797 |
| CONDITIONAL ON STATIC SIGNAL | | | |
| POSITIONAL | 10.1 | 41.1% | 1,755 |
| TOKEN EMBEDDING | 10.0 | 31.1% | 1,966 |

strong sink behavior. Among the conditional gating configurations, per-channel and per-head gates both eliminate attention sinks entirely. A single gate per token, however, yields elevated sink ratios and slightly higher perplexity—consistent with our earlier finding that sink formation is a head-level phenomenon. Under unconditional gating, the static gate fails to suppress either attention sinks or massive activations. Similarly, gating conditioned on positional or token embeddings does not eliminate sinks, as these signals are fixed and cannot adapt to the evolving context.

Taken together, these results suggest that attention sinks serve as a form of implicit input-conditioned gating: the effective routing behavior depends on the prompt history rather than being a fixed property of a particular head, position, or token. When the model has access to a dynamic, representation-conditioned gate, it can modulate attention routing on the fly, eliminating the structural need to maintain a spike token via large residual spikes.

### 3.3.3. TRAINING CONTEXT LENGTH

Xiao et al. (2024a) suggest that attention sinks facilitate short-range dependence in sink heads. Consistent with this, we observe that sink heads predominantly attend to nearby tokens of the query. We therefore vary the training context-length distribution to test whether sinks are an inductive bias of short-range training, controlling the distribution by adjusting the range of sequence positions over which the training loss is computed. Results are shown in Table 8.

When the training distribution includes short sequences, the sink ratio remains stable regardless of the maximum context length. Removing short contexts entirely—optimizing only over long-range positions—causes the sink ratio to collapse dramatically. This confirms that attention sinks are funda-

*Table 8.* **Context-length ablations.** Attention sinks are largely induced to facilitate short-context prediction. When the training distribution is restricted to long sequences, the sink ratio collapses, indicating that sinks are primarily utilized to support short-range dependence. Highlighted rows denote the baseline configuration.

| SETUP | PERPLEXITY | SINK RATIO | SPIKE |
|---|---|---|---|
| CONTEXT LENGTH (MIN/MAX) | | | |
| 1/256 | 12.4 | 42.1% | 5,411 |
| 1/1024 | 10.6 | 46.3% | 4,442 |
| 1/4096 | 10.1 | 46.0% | 3,818 |
| 1024/4096 | 10.1 | 13.0% | 38,470 |
| 1024/5120 | 10.1 | 8.0% | 42,365 |
| 2048/4096 | 10.6 | 1.2% | 7,193 |
| 2048/6144 | 10.0 | 5.8% | 30,634 |

mentally a byproduct of short-context training: in mixed-length regimes, the first token provides a cheap, universally available global reference that to reduce the influence of far away tokens. Excluding short-context positions from the training loss therefore reveals that the majority of sinks are induced specifically to facilitate local prediction within a global attention mechanism, corroborating the role of sink heads identified by Xiao et al. (2024a).

### 3.4. Summary and Discussion

Our ablation study reveals three critical insights into massive activations and attention sinks:

1. **Causal Independence of Spikes.** While spikes and sinks often co-occur, they are not inextricably linked. Normalization techniques such as Sandwich Norm, Dynamic-Tanh, and QKNorm can eliminate massive activations without destroying attention sinks. This suggests that spikes are an artifact of the Pre-Norm architecture's tendency to accumulate unbounded values, which the model can exploit—but does not require—for sink formation.

2. **Sinks as a Gating Mechanism.** The disappearance of sinks under Conditional Gating suggests that attention sinks act as a learned workaround for modulating information flow. In the absence of an explicit gate, the model repurposes the first token as a numerical "dumping ground" to gate off unnecessary attention heads. Once an explicit dynamic gate is provided, this implicit routing mechanism becomes unnecessary.

3. **Context-Length Induction.** Sinks are also driven by the need to model short-range dependencies with a global attention mechanism. By assigning attention to the first token, the model can suppress irrelevant long-range context when it is not predictive. This behavior becomes less necessary when the model is trained exclusively on long-context sequences.

Together, these results show that massive activations and attention sinks can be independently controlled. Their frequent co-occurrence in standard LLMs reflects incidental architectural interactions, especially through normalization, rather than a necessary functional coupling.

## 4. Relation to Prior Work

Due to page limits, we delay the comprehensive related work to Appendix E. Here, we present a detailed account of some of our findings' relation to prior work.

Several findings we analyze in Sections 2 and 3 were documented in isolation by prior work; we unify and extend them into a single mechanistic account, sharpening each with emphasis on the cause. We make this lineage explicit below, stating for each finding the established prior work and our extended work. Sun et al. (2024) characterize when and where massive activations occur, we distill their behavior into five properties and trace their initial and final emergence to step-up and step-down blocks in early and late layers. While Yu et al. (2024a); An et al. (2025) attribute massive activations to FFN weight outliers, with weight alignment further driving activation spikes (Yu et al., 2024a; Fishman et al., 2025; Yang et al., 2024a), we show that the SwiGLU FFN acts as a directional quadratic amplifier, establishing rank-one dominance in the residual stream. We link the first-token phenomenon (Sun et al., 2024) to a static linear transformation at initial positions. Prior work observes query/key geometric alignment (Gu et al., 2025; Devoto et al., 2024; Guo et al., 2024); we identify its driving factor as the normalized sparse token representation, which forces sink-token keys into a low-dimensional subspace. Our training-time hyperparameter ablations confirm Gu et al. (2025) that sinks are less pronounced under small learning rates while weight decay encourages them, and corroborate Owen et al. (2025a) that massive activations vanish when normalization is replaced by pointwise functions such as DynamicTanh. Finally, extending the gating-based outlier reduction of Qiu et al. (2025), we find that attention sinks act as a compensatory mechanism, emerging precisely when conditional gating is unavailable.

## 5. Conclusion

This research clarifies the relationship between massive activations and attention sinks in large language models. The findings demonstrate that these phenomena, while often co-occurring, are not inherently linked but are instead decoupled architectural artifacts of the pre-norm Transformer design. By identifying the specific roles of normalization and residual accumulation, the study shows that massive activations function as global implicit parameters, whereas attention sinks serve as local modulators for attention heads.

The evidence suggests that both phenomena can be independently mitigated through alternative architectural choices. This separation of functional roles provides a clearer path for future model optimization, particularly for improving quantization, pruning, and long-context inference. Ultimately, these insights move beyond descriptive observations to offer a structural understanding of how internal representations are shaped by specific training and design decisions.

## Impact Statement

This work advances the understanding of internal mechanisms in Transformer-based language models, particularly regarding extreme activation patterns and attention concentration phenomena. Improved interpretability of large models can contribute to more reliable, efficient, and controllable AI systems, with potential benefits for deployment in scientific, educational, and industrial applications.

We do not anticipate direct negative societal impacts arising from this work. The methods focus on analysis and architectural design rather than new capabilities for content generation or decision-making, and primarily support safer and more transparent model development.

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

# A. Preliminaries

Modern LLMs are typically trained on the next-token prediction task (Bengio et al., 2003; Radford et al., 2018) over large text corpora, using decoder-only Transformers (Vaswani et al., 2017) with the pre-norm configuration (Xiong et al., 2020). Despite variations in model size and training data, these choices have remained remarkably consistent across most major models. This section formalizes these core components and establishes the notation used throughout subsequent discussion.

## A.1. Next-Token Prediction

Next-token prediction is a self-supervised learning objective that leverages the sequential structure of natural language. By treating token order as a natural supervisory signal, models can be trained on vast unlabeled corpora. Formally, let $\mathbf{x} := (x_1, \ldots, x_T)$ be a sequence of T tokens, where each token $x_i$ takes values in a finite vocabulary $\mathcal{V}$. A language model parameterized by $\theta$ defines a joint distribution:

$$\mathbb{P}_\theta(\mathbf{x}) = \mathbb{P}_\theta(x_1, \ldots, x_T). \tag{14}$$

Direct modeling of this joint distribution is computationally intractable due to the exponential growth of the sample space. Autoregressive models address this by factorizing the joint distribution into a product of conditional probabilities:

$$\mathbb{P}_\theta(x_1, \ldots, x_T) = \prod_i \mathbb{P}_\theta(x_i \mid \mathbf{x}_{<i}), \tag{15}$$

where $\mathbf{x}_{<i} := (x_1, \ldots, x_{i-1})$ represents the prefix (context) preceding index $i$.

In decoder-only Transformers, each conditional is computed by mapping the prefix $\mathbf{x}_{<i}$ to a distribution over $\mathcal{V}$. During training, all conditionals are produced in parallel by supplying the ground-truth prefix at every position via teacher forcing (Williams & Zipser, 1989). Given a training corpus $\mathcal{D}$, parameters $\theta$ are learned by minimizing the expected negative log-likelihood:

$$\mathcal{L}(\theta) := -\mathbb{E}_{\mathbf{x} \sim \mathcal{D}} \left[ \sum_i \log \mathbb{P}_\theta(x_i \mid \mathbf{x}_{<i}) \right]. \tag{16}$$

This objective reduces language modeling to a sequence of conditional classification problems over $\mathcal{V}$, with the conditioning context growing with $i$.

## A.2. Transformer Architecture

Since the introduction of the Transformer model, many architectural variants have been proposed, and modern LLMs differ in numerous details. In this section, we describe the specific architecture used by the Llama family of LLMs (Touvron et al., 2023a;b; Grattafiori et al., 2024). We focus on Llama because it is among the most widely used open-weight models, and its design choices have strongly influenced subsequent open models such as Qwen (Qwen et al., 2025; Yang et al., 2025) and Mistral (Liu et al., 2026).

**Token embedding.** A natural language sentence is first decomposed into a sequence of discrete tokens by a tokenizer, then mapped to continuous vectors via an embedding table. Specifically, each token is mapped to a $d_{model}$-dimensional vector. For a sequence of T tokens, we denote the resulting hidden representation by $\mathbf{H}_1 \in \mathbb{R}^{T \times d_{model}}$.

**Transformer layers.** Starting from $\mathbf{H}_1$, a stack of L Transformer layers transforms the hidden representation while preserving its dimensionality. Each layer consists of two blocks—an attention block and a feed-forward block—yielding 2L blocks in total.

Let $\mathbf{H}_i \in \mathbb{R}^{T \times d_{model}}$ denote the input to block $i$, and let $\mathcal{F}_i(\cdot)$ denote its transformation. Every block employs a residual connection with pre-norm configuration:

$$\mathbf{H}_{i+1} = \mathbf{H}_i + \mathcal{F}_i(\text{RMSNorm}(\mathbf{H}_i)), \tag{17}$$

where $\mathcal{F}_i$ is the attention block when $i$ is odd and the feed-forward block when $i$ is even. The function $\text{RMSNorm}(\cdot)$ (Zhang & Sennrich, 2019) is applied row-wise:

$$\text{RMSNorm}(\mathbf{h}) := \sqrt{d_{model}} \frac{\mathbf{h}}{\|\mathbf{h}\|}, \tag{18}$$

where $\mathbf{h} \in \mathbb{R}^{d_{\text{model}}}$ is a single row of $\mathbf{H}_i$. We omit the learnable scale parameter from the RMSNorm formulation here, since every RMSNorm is immediately followed by a linear layer and the scale parameter can be absorbed into the subsequent weight matrix during the forward pass.

**Attention block.** The attention mechanism is implemented as multi-head attention with $N_{\text{head}}$ heads, each of dimension $d_{\text{head}}$. For each head $i$, the normalized input $\tilde{\mathbf{H}} := \text{RMSNorm}(\mathbf{H})$ is projected using head-specific weight matrices $\mathbf{W}_Q^{(i)}, \mathbf{W}_K^{(i)}, \mathbf{W}_V^{(i)} \in \mathbb{R}^{d_{\text{model}} \times d_{\text{head}}}$:

$$\mathbf{Q}^{(i)} := \tilde{\mathbf{H}}\mathbf{W}_Q^{(i)}, \tag{19}$$

$$\mathbf{K}^{(i)} := \tilde{\mathbf{H}}\mathbf{W}_K^{(i)}, \tag{20}$$

$$\mathbf{V}^{(i)} := \tilde{\mathbf{H}}\mathbf{W}_V^{(i)}, \tag{21}$$

$$\mathbf{A}^{(i)} := \text{softmax}\left( \frac{\mathbf{Q}^{(i)}\mathbf{K}^{(i)\top}}{\sqrt{d_{\text{head}}}} + \mathbf{M}_{\text{causal}} \right), \tag{22}$$

$$\mathbf{O}^{(i)} := \mathbf{A}^{(i)}\mathbf{V}^{(i)}, \tag{23}$$

where $\text{softmax}$ is applied row-wise to ensure each row of $\mathbf{A}^{(i)}$ forms a valid probability distribution. The causal mask $\mathbf{M}_{\text{causal}} \in \mathbb{R}^{T \times T}$ enforces the autoregressive property: its entries are 0 on and below the diagonal and $-\infty$ above, preventing each position from attending to future tokens. For simplicity, we omit positional encoding from our description. In practice, Llama applies Rotary Position Embeddings (Su et al., 2024) to the $\mathbf{Q}^{(i)}$ and $\mathbf{K}^{(i)}$ before computing $\mathbf{A}^{(i)}$.

The per-head outputs are concatenated and projected via $\mathbf{W}_O \in \mathbb{R}^{(N_{\text{head}} \cdot d_{\text{head}}) \times d_{\text{model}}}$ to produce the final output:

$$\mathcal{F}_{\text{attn}}(\tilde{\mathbf{H}}) := \text{Concat}\left( \mathbf{O}^{(1)}, \ldots, \mathbf{O}^{(N_{\text{head}})} \right) \mathbf{W}_O. \tag{24}$$

**Feed-forward block.** While the attention block facilitates information exchange across token positions, the feed-forward block operates independently on each position. Modern LLMs typically employ the SwiGLU activation function (Shazeer, 2020). For an input vector $\tilde{\mathbf{h}} \in \mathbb{R}^{d_{\text{model}}}$ (a row of $\tilde{\mathbf{H}}$), the feed-forward transformation is defined as:

$$\mathcal{F}_{\text{ffn}}(\tilde{\mathbf{h}}) := \mathbf{W}_{\text{down}} \cdot \left( \text{SiLU}(\mathbf{W}_{\text{gate}}\tilde{\mathbf{h}}) \odot (\mathbf{W}_{\text{up}}\tilde{\mathbf{h}}) \right), \tag{25}$$

where $\odot$ denotes the element-wise (Hadamard) product. The weight matrices are the gate-projection $\mathbf{W}_{\text{gate}} \in \mathbb{R}^{d_{\text{ffn}} \times d_{\text{model}}}$, the up-projection $\mathbf{W}_{\text{up}} \in \mathbb{R}^{d_{\text{ffn}} \times d_{\text{model}}}$, and the down-projection $\mathbf{W}_{\text{down}} \in \mathbb{R}^{d_{\text{model}} \times d_{\text{ffn}}}$. Here $d_{\text{ffn}}$ denotes the intermediate dimension, which is typically three or four times large than $d_{\text{model}}$.

**Prediction head.** After all $2L$ blocks, the final hidden representation passes through a RMSNorm layer and a linear projection to produce logits for next-token prediction:

$$\mathbf{Y} := \text{RMSNorm}(\mathbf{H}_{2L+1}) \mathbf{W}_{\text{head}}, \tag{26}$$

where $\mathbf{H}_{2L+1}$ is the output of the last residual block, $\mathbf{W}_{\text{head}} \in \mathbb{R}^{d_{\text{model}} \times |\mathcal{V}|}$ is the projection head, and $\mathbf{Y} \in \mathbb{R}^{T \times |\mathcal{V}|}$ is the matrix of output logits.

# B. Implementation Details

All models in Section 3 are trained on the DCLM dataset (Li et al., 2024) using a shared codebase and a common baseline recipe. Unless otherwise noted, we keep the training pipeline fixed and vary only the factors under study in each ablation. The default recipe closely follows the Llama-style pretraining setup (Touvron et al., 2023a). With some additional details that we couldn't find from the original paper, we refer to the *torchtitan* (Liang et al., 2025b) and *Olmo* (Olmo et al., 2025) codebase. Tables 9 report the baseline architecture and optimization hyperparameters used across experiments.

**Datasets.** During evaluation, we randomly sample text from the C4 corpus (Raffel et al., 2020), with a total budget of up to $1024 \times 4096$ tokens. We tokenize the sampled text and partition it into fixed-length chunks of $\{64, 256, 1024, 2048, 4096\}$ tokens, selecting the chunk size to match the configured context window for each model.

*Table 9.* **Baseline training configurations.**

| *(a)* Model architecture. | | | *(b)* Optimization configurations. | |
|---|---|---|---|---|
| **Hyperparameter** | **Value** | | **Hyperparameter** | **Value** |
| Architecture | Llama 2 | | Optimizer | AdamW |
| Parameters | $\sim 6.7B$ | | $\beta_1, \beta_2$ | 0.9, 0.95 |
| Layers | 32 | | Weight decay | 0.1 |
| Hidden size | 4,096 | | Gradient clipping | 1.0 |
| Attention heads | 32 | | Base learning rate | $3.0 \times 10^{-4}$ |
| Head dimension | 128 | | LR schedule | Cosine decay |
| Intermediate size | 11,008 | | Warmup schedule | Quadratic |
| Vocabulary size | 32,000 | | Warmup steps | 2,000 |
| Parallelism | FSDP2 | | Training steps | 5,000 |
| Precision | BF16 | | Batch size | 2M **tokens** |
| Compilation | `torch.compile` | | Min LR ratio | 0.1 |

**Sink Ratio.** For an input sequence of length $T$, let $A^{l,h} \in [0,1]^{T \times T}$ denote the (causal) self-attention matrix at layer $l$ and head $h$, where $A^{l,h}_{t,k}$ is the attention weight from query position $t$ to key position $k$. Following Gu et al. (2025), we define the *importance score* of position $k$ as the attention it receives on average:

$$\alpha^{l,h}_k \; := \; \frac{1}{T} \sum_{t=1}^{T} A^{l,h}_{t,k}. \tag{27}$$

A head is said to exhibit an attention sink (at threshold $\epsilon$) if there exists a position in the first half of the sequence that receives more than $\epsilon$ average attention, i.e., $\max_{1 \leq k \leq T/2} \alpha^{l,h}_k > \epsilon$. We then define the *sink ratio* for this sequence as the fraction of heads (averaged over heads across all layers) that satisfy this criterion:

$$s_\epsilon \; = \; \frac{1}{LH} \sum_{l=1}^{L} \sum_{h=1}^{H} \mathbf{1}\left( \max_{1 \leq k \leq [T/2]} \alpha^{l,h}_k > \epsilon \right). \tag{28}$$

Finally, we report the model-level sink ratio by averaging $s_\epsilon$ over evaluation sequences. In our experiments, we use $\epsilon = 0.3$ and $T = 64$ consistently.

## C. Theorems and Derivations

**Theorem C.1 (Attention output as a sum over heads).** *Let* $\mathbf{O}^{(h)} := \mathbf{A}^{(h)} \mathbf{V}^{(h)} \in \mathbb{R}^{T \times d_{\text{head}}}$ *denote the output of head* $h$ *with* $\mathbf{V}^{(h)} := \tilde{\mathbf{H}} \mathbf{W}_V^{(h)}$ *and* $\tilde{\mathbf{H}}$ *being the hidden representations of inputs right before attention block. Let* $\mathbf{W}_O \in \mathbb{R}^{(N_{\text{head}} \cdot d_{\text{head}}) \times d_{\text{model}}}$ *be the output projection weight matrix and partition it by head as*

$$\mathbf{W}_O = \text{Concat}\left( \mathbf{W}_O^{(1)}, \ldots, \mathbf{W}_O^{(N_{\text{head}})} \right)^\top \tag{29}$$

*with* $\mathbf{W}_O^{(h)} \in \mathbb{R}^{d_{\text{head}} \times d_{\text{model}}}$, $h \in \{1, \ldots, N_{\text{head}}\}$.

*Then the attention block output*

$$\text{Concat}\left( \mathbf{O}^{(1)}, \ldots, \mathbf{O}^{(N_{\text{head}})} \right) \cdot \mathbf{W}_O \tag{30}$$

*can be written as*

$$\sum_{h=1}^{N_{\text{head}}} \mathbf{A}^{(h)} \tilde{\mathbf{H}} \mathbf{W}_V^{(h)} \mathbf{W}_O^{(h)}. \tag{31}$$

*Proof.* By definition of each head,

$$\mathbf{O}^{(h)} = \mathbf{A}^{(h)} \mathbf{V}^{(h)} = \mathbf{A}^{(h)} \tilde{\mathbf{H}} \mathbf{W}_V^{(h)}. \tag{32}$$

Using the head-wise block partition of $\mathbf{W}_O$, multiplying the concatenation by $\mathbf{W}_O$ decomposes additively:

$$\text{Concat}(\mathbf{O}^{(1)}, \ldots, \mathbf{O}^{(N_{\text{head}})}) \cdot \mathbf{W}_O = \sum_{h=1}^{N_{\text{head}}} \mathbf{O}^{(h)} \mathbf{W}_O^{(h)}. \tag{33}$$

Substituting $\mathbf{O}^{(h)} = \mathbf{A}^{(h)} \tilde{\mathbf{H}} \mathbf{W}_V^{(h)}$ into the above equation yields

$$\text{Concat}\left(\mathbf{O}^{(1)}, \ldots, \mathbf{O}^{(N_{\text{head}})}\right) \mathbf{W}_O = \sum_{h=1}^{N_{\text{head}}} \mathbf{A}^{(h)} \tilde{\mathbf{H}} \mathbf{W}_V^{(h)} \mathbf{W}_O^{(h)}. \tag{34}$$

$\square$

**Theorem C.2** (**Quadratic-form approximation of a SiLU feed-forward coordinate**). *Under the approximation of SiLU function (Equation* (2)*) for spike tokens with input representations* $\tilde{\mathbf{h}}$*, the output of the feed-forward block* $\mathbf{y}$ *has the quadratic form approximation for coordinate* $k$ *as follows*

$$\mathbf{y}_k = \tilde{\mathbf{h}}^\top \mathbf{U}_k \tilde{\mathbf{h}}, \tag{35}$$

*where*

$$\mathbf{U}_k := \sum_{i=1}^{d_{\text{ff}}} \mathbf{W}_{\text{down}}^{(k,i)} \mathbf{W}_{\text{gate}}^{(i)} \mathbf{W}_{\text{up}}^{(i)\top}. \tag{36}$$

*Proof.* Let $\mathbf{z} \in \mathbb{R}^{d_{\text{hidden}}}$ denote the intermediate hidden state

$$\mathbf{z} := \mathbf{W}_{\text{gate}} \tilde{\mathbf{h}} \odot \mathbf{W}_{\text{up}} \tilde{\mathbf{h}}. \tag{37}$$

The $i$-th element of $\mathbf{z}$, i.e. $\mathbf{z}_i$ could be represented as

$$\begin{aligned} \mathbf{z}_i &= (\mathbf{W}_{\text{gate}}^{(i)\top} \tilde{\mathbf{h}}) \cdot (\mathbf{W}_{\text{up}}^{(i)\top} \tilde{\mathbf{h}}) \\ &= \tilde{\mathbf{h}}^\top \mathbf{W}_{\text{gate}}^{(i)} \mathbf{W}_{\text{up}}^{(i)\top} \tilde{\mathbf{h}}, \end{aligned} \tag{38}$$

where $\mathbf{W}_{\text{gate}}^{(i)}$ and $\mathbf{W}_{\text{up}}^{(i)}$ are the $i$-th rows of the weight matrices. The term $\mathbf{W}_{\text{gate}}^{(i)} \mathbf{W}_{\text{up}}^{(i)\top}$ is equal to the outer product of two vectors, forming a rank-1 matrix $\mathbf{V}_i$. We can now rewrite the hidden element as the quadratic form

$$\mathbf{z}_i = \tilde{\mathbf{h}}^\top \mathbf{V}_i \tilde{\mathbf{h}}. \tag{39}$$

Next, consider the $k$-th element of the output vector $\mathbf{y}$, denoted $\mathbf{y}_k$. By the linearity of the final projection ($\mathbf{W}_{\text{down}}$):

$$\begin{aligned} \mathbf{y}_k &= \sum_i \mathbf{W}_{\text{down}}^{(k,i)} \mathbf{z}_i \\ &= \sum_i \mathbf{W}_{\text{down}}^{(k,i)} \cdot \left(\tilde{\mathbf{h}}^\top \mathbf{V}_i \tilde{\mathbf{h}}\right) \\ &= \tilde{\mathbf{h}}^\top \left(\sum_i \mathbf{W}_{\text{down}}^{(k,i)} \mathbf{V}_i\right) \tilde{\mathbf{h}}. \end{aligned} \tag{40}$$

Therefore, we conclude that the output element is in quadratic form

$$\mathbf{y}_k = \tilde{\mathbf{h}}^\top \mathbf{U}_k \tilde{\mathbf{h}}, \tag{41}$$

where the matrix $\mathbf{U}_k$ is defined as the weighted sum of the rank-1 components

$$\mathbf{U}_k = \sum_i \mathbf{W}_{\text{down}}^{(k,i)} \mathbf{V}_i = \sum_i \mathbf{W}_{\text{down}}^{(k,i)} \cdot \left(\mathbf{W}_{\text{gate}}^{(i)} \mathbf{W}_{\text{up}}^{(i)\top}\right). \tag{42}$$

This derivation establishes the form given in Equation (4) and concludes the proof. $\square$

**Theorem C.3** (**Coordinate bound under RMS normalization**). *Let* $\mathbf{h} \in \mathbb{R}^{d_{\text{model}}}$ *be any non-zero vector and define its RMS-normalized version*

$$\tilde{\mathbf{h}} \;:=\; \text{RMSNorm}(\mathbf{h}) \;=\; \sqrt{d_{\text{model}}}\,\frac{\mathbf{h}}{\|\mathbf{h}\|_2}. \tag{43}$$

*Then every coordinate of* $\tilde{\mathbf{h}}$ *is bounded in magnitude by* $\sqrt{d_{\text{model}}}$:

$$\left|\tilde{\mathbf{h}}_i\right| \;\leq\; \sqrt{d_{\text{model}}}, \quad \forall i \in \{1, \ldots, d_{\text{model}}\}. \tag{44}$$

*Proof.* For any coordinate $i \in \{1, \ldots, d_{\text{model}}\}$, by definition,

$$\left|\tilde{\mathbf{h}}_i\right| = \sqrt{d_{\text{model}}}\,\frac{|\mathbf{h}_i|}{\|\mathbf{h}\|_2}. \tag{45}$$

Since

$$\|\mathbf{h}\|_2^2 = \sum_{j=1}^{d_{\text{model}}} \mathbf{h}_j^2 \geq \mathbf{h}_i^2, \tag{46}$$

we have $\|\mathbf{h}\|_2 \geq |\mathbf{h}_i|$. Therefore,

$$\frac{|\mathbf{h}_i|}{\|\mathbf{h}\|_2} \leq 1 \quad \implies \quad \left|\tilde{\mathbf{h}}_i\right| \leq \sqrt{d_{\text{model}}}. \tag{47}$$

$\square$

# D. Additional Results

**Open-Source Models.** While our primary analysis focused on Llama 2 7B, we validate the universality of our findings across the diverse set of open-source models detailed in Table 10. These models span multiple families (Llama 2, Llama 3, Qwen2.5, Qwen3), depths (28 to 48 layers), and parameter counts (7B to 14B).

*Table 10.* **List of open source models evaluated in Appendix D.** We observe consistent massive activations and attention sinks phenomena across the following diverse model families and sizes.

| Model Family | Model Name | Layers | Dimensions | Heads | Huggingface Model Id |
|---|---|---|---|---|---|
| LLAMA | Llama 2 7B | 32 | 4096 | 32 | `meta-llama/Llama-2-7b-hf` |
| | Llama 2 13B | 40 | 5120 | 40 | `meta-llama/Llama-2-13b-hf` |
| | Llama 3 8B | 32 | 4096 | 32 | `meta-llama/Meta-Llama-3-8B` |
| QWEN | Qwen2.5 7B | 28 | 3584 | 28 | `Qwen/Qwen2.5-7B` |
| | Qwen2.5 14B | 48 | 5120 | 40 | `Qwen/Qwen2.5-14B` |
| | Qwen3 8B | 36 | 4096 | 32 | `Qwen/Qwen3-8B` |
| | Qwen3 14B | 40 | 5120 | 40 | `Qwen/Qwen3-14B` |

**Universality of Step-Up/Step-Down Dynamics.** Figure 7 visualizes the top-3 coordinate magnitudes through the residual stream for all 12 evaluated models. We observe two consistent behaviors across all architectures:

1. **Massive Activations:** Every model exhibits activation spikes orders of magnitude larger than the baseline variance.
2. **Feed Forward Block Driven Origin:** Comparing the "post-residuals" (top panels) and "block outputs" (bottom panels) plots in Figure 7 confirms that these spikes are not merely accumulated residual error. Instead, they originate abruptly at specific "step-up" blocks and are neutralized by subsequent "step-down" blocks, matching the mechanism described in Section 2.

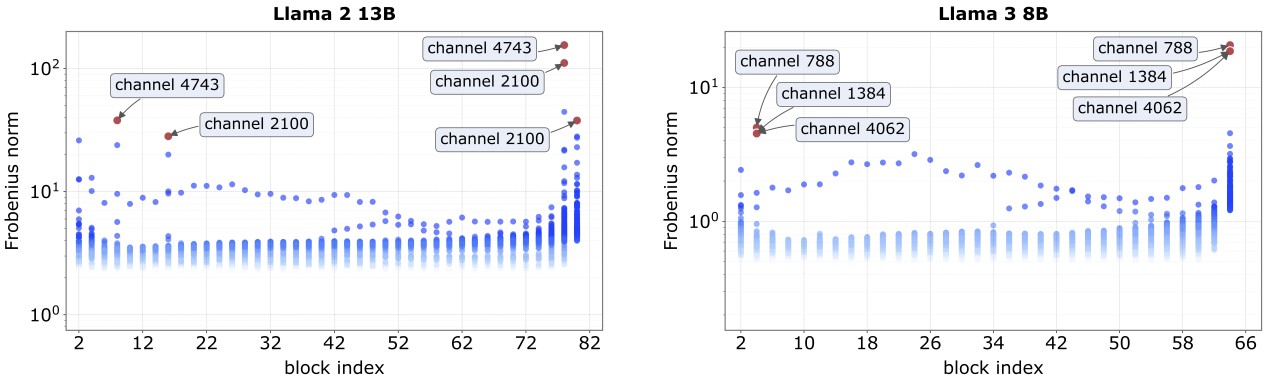

*Figure 7.* **Top-3 coordinate magnitudes after (top panels) and before (bottom panels) the residual branch for 12 open source models.** All models exhibit the characteristic step-up and step-down behavior driven by one or few early blocks and then the late blocks.

**Universality of Frobenius norm outliers.** To confirm that these activations arise from the directional quadratic amplification mechanism, we analyze the Frobenius norms of the quadratic form matrices $\mathbf{U}_k$. As shown in Figure 8, channels exhibiting massive activations correspond to $\mathbf{U}_k$ matrices with exceptionally large Frobenius norms.

For example, in Llama 3 8B, distinct spikes are visible at channels 788, 1384, and 4062, which align with the massive activation spikes observed in Figure 7. This confirms that the alignment of attention sinks with high-gain quadratic directions in the MLP is a structural invariant across Llama model families.

*Figure 8.* **Frobenius norms $\|\mathbf{U}_k\|_F$ for the quadratic forms of Llama models.** Spike channels align with $\mathbf{U}_k$ matrices that exhibit substantially larger norms than typical channels. These high-norm coordinates appear exclusively in step-up and step-down blocks.

# E. Additional Related Work

**Attention sinks** have been observed across various Transformer models of all sizes (Xiao et al., 2024b; Yu et al., 2024b), including LLMs, vision-language models (Zhang et al., 2026), and multimodal models (Cappellazzo et al., 2025). Early evidence of this phenomenon traces back to the identification of outlier dimensions in BERT-scale models (Kovaleva et al., 2021), with their emergence linked to the frequency of tokens in training data (Puccetti et al., 2022). They are not considered a downstream-task artifact (Owen et al., 2025a): they emerge during pre-training and persist through instruction tuning (Gu et al., 2025; Zhai et al., 2023; Wortsman et al., 2023). Proposed explanations span multiple hypotheses (Liu et al., 2024; Sandoval-Segura et al., 2026; Zhang et al., 2025; Shang et al., 2025), with many attributing attention sinks to softmax normalization in self-attention (Ali et al., 2025; An et al., 2025; Guo et al., 2024; Lin et al., 2025; Miller, 2023; Noci et al., 2023; Veličković et al., 2024; Saada et al., 2024; Xiao et al., 2024b).

Recent work has characterized the functional role of sinks, identifying "dormant" (Sandoval-Segura et al., 2026) or "garbage" (Sok et al., 2026) attention heads dominated by sink tokens that act as redundant dumping grounds. This structure has been exploited for efficient inference through "streaming" heads (Xiao et al., 2024a), adaptive KV cache eviction that preserves first tokens (Ge et al., 2024), layer-condensed cache strategies (Wu & Tu, 2024), and hybrid sparse attention patterns (Fu et al., 2025). However, the disproportionate attention captured by boundary sinks also leads to low information retrieval for the middle part of long contexts (Liu et al., 2023).

**Massive activations.** They were first identified in LLMs as extreme outlier features concentrated in specific channels (Dettmers et al., 2022) and have been systematically characterized and shown to co-locate with attention sink tokens later (Sun et al., 2024). These outliers start to emerge during pre-training and are strongly tied to architecture choices such as pre-normalization (He et al., 2024); they become increasingly pronounced as models scale (Ahmadian et al., 2023). They display behavior analogous to implicit bias terms: their magnitudes are stable across inputs, and they are tightly coupled with "massive weights" whose perturbation causes performance collapse (Oh et al., 2024). These activations occupy fixed, largely input-agnostic dimensions and can be induced by highly aligned "super" weights (Yu et al., 2024a).

The presence of massive activations poses significant challenges for low-precision serving and training. Outlier channels severely degrade quantization performance (Wei et al., 2022; Bondarenko et al., 2021), necessitating specialized techniques such as per-token scaling (Yao et al., 2022), mixed-precision decomposition (Dettmers et al., 2022; Zhao et al., 2024; Huang et al., 2024), and outlier migration via shifting or Hadamard transformations (Wei et al., 2023; Xi et al., 2023; Wang et al., 2025). Furthermore, these activations can amplify numerical rounding errors during inference (Budzinskiy et al., 2025) and complicate ultra-low-precision FP4 training (Abecassis et al., 2025).

**Mitigations and unifying theories** that jointly explain these phenomena have recently emerged. Training-time mitigations primarily modify the attention mechanism or normalization layers. Alternatives to softmax include *sigmoid* (Gu et al., 2025; Ramapuram et al., 2024), *ReLU* (Guo et al., 2024), *softmax-off-by-one* (Kaul et al., 2024; Miller, 2023), and Elastic-Softmax (Fu et al., 2026). Theoretical analysis of LayerNorm suggests it drives outlier emergence through re-centering and re-scaling (Xu et al., 2019); consequently, normalization-free architectures (Chen et al., 2025) and "Outlier Protected" blocks (He et al., 2024) have been proposed to eliminate the structural incentive for sinks.

Proactive strategies to suppress outliers include spectral sphere constraints (Xie et al., 2026), smooth architectural modifications (Liang et al., 2025a), and explicit weight rescaling during pre-training (Owen et al., 2025b). Alternative routing mechanisms, such as hybrid attention-SSM architectures (Dong et al., 2024) or gated networks (Yang et al., 2024b), provide explicit capacity control that reduces reliance on sinks. These findings complement theories like Mix-Compress-Refine (Queipo-de Llano et al., 2025), which posits that massive activations drive the "compression valley" (Skean et al., 2025) and induce spectral dominance. Joint emergence has also been attributed to adaptive optimization dynamics, motivating orthogonalized variants such as OrthoAdam (Kaul et al., 2024).

# F. Limitations and Future Work

While our investigation provides a unified mechanistic framework linking massive activations and attention sinks in pre-norm decoder-only Transformers, we acknowledge several boundaries to our current methodology. In this section, we detail these limitations and outline concrete directions for future research, particularly addressing areas where our static, inference-time analyses can be expanded into dynamic, causal, and optimization-centric evaluations.

**Direct Causal Interventions** A primary limitation of our current study is the reliance on observational and correlational analysis during the forward pass. Although our controlled architectural ablations strongly imply a causal chain—from step-up feed-forward blocks to sparse normalized representations and subsequent attention sink formation—we do not execute direct, inference-time interventions on frozen models. Future work should implement targeted causal interventions to definitively validate this pathway. Specifically, we propose experiments that explicitly clamp the top-$k$ outlier channels to their mean values at the exact step-up layer, or project out the primary high-gain eigenvector $s_*$ from the SwiGLU output. Preliminary hypotheses suggest that surgically neutralizing these massive activations will disrupt the geometric segregation of sink keys $k^{(s)}$ and non-sink keys $k^{(n)}$, thereby dissolving the disproportionate logit gaps that sustain attention sinks. Such direct interventions would complement our architectural ablations and provide a rigorous causal proof of the inference-time mechanism.

**Downstream Task Evaluation** Our ablation studies (Section 3 and 4) evaluate model configurations predominantly through the lenses of validation perplexity and the attention sink ratio. While perplexity is an established proxy for general optimization health and language modeling capability, it does not fully capture the nuanced impacts of suppressing attention sinks or massive activations on practical downstream tasks. It remains an open question whether the absence of attention sinks—achieved through techniques like dynamic gating or restricted minimum context lengths—adversely affects capabilities such as zero-shot reasoning, few-shot in-context learning, or long-context retrieval (e.g., "needle-in-a-haystack" evaluations). Future investigations must extend beyond cross-entropy loss to evaluate sink-free and spike-free architectures on comprehensive benchmarks like MMLU, LongBench, and standard generative tasks to ensure that mitigating these phenomena does not introduce hidden trade-offs in generation quality or context utilization.

**Optimization-Time Dynamics** The scope of our work is largely centered on the inference-time mechanics of pre-trained networks. While we demonstrate that hyperparameter choices (such as learning rate and weight decay) modulate the intensity of these phenomena, we do not present a comprehensive optimization-time theory that explains *why* the network learns to align weight matrices $W_{\text{gate}}$ and $W_{\text{up}}$ into collinear, high-gain directions in the first place. Preliminary checkpoint analyses indicate that massive activations and attention sinks manifest extremely early in the pre-training process (e.g., within the first 4,000 steps) and gradually amplify as training progresses. Future research should investigate the gradient dynamics and adaptive optimizer states (e.g., AdamW momentum) that drive this early spectral dominance, bridging our forward-pass amplification theory with a formal understanding of the pre-training trajectory.

**Scale, Training Budget, and Seed Robustness** To conduct controlled, from-scratch ablations, our experimental setup was constrained to 7B-parameter models trained on a 100B token budget. Although we verified the existence of massive activations and attention sinks across a broad suite of pre-trained, open-weight models ranging up to 72B parameters, the generalizability of our specific causal ablations to massive scales remains partially unverified. Future work must rigorously test these architectural modifications (e.g., DynamicTanh, minimum context length constraints) across varying random seeds, significantly larger parameter scales, and extended training budgets (e.g., $> 1$T tokens). Confirming that these mitigations remain stable and do not degrade under the immense gradient noise of scale will be critical for integrating them into state-of-the-art LLM pre-training pipelines.

**Granularity of Linear Block Accumulation** Finally, our ablation involving the replacement of SwiGLU blocks with standard linear feed-forward networks revealed that outliers still emerge, albeit through a more diffuse accumulation across multiple layers rather than a sharp, single-step injection. The current manuscript lacks a detailed visual and quantitative analysis of this gradual accumulation. Future extensions of this work will include layer-by-layer tracking metrics and Frobenius norm analyses of the linear block weight matrices to visually contrast the sudden rank-one amplification of SwiGLU with the slow residual buildup in linear architectures. This granular view will further illuminate how the residual stream serves as an additive conduit for implicit parameters, regardless of the underlying nonlinearity.

