# OpenReview forum: "Anatomy of Massive Activations and Attention Sinks"
_ICML.cc/2026/Conference — ICML 2026 regular_

### Official Review · Reviewer_NJUF · 2026-03-03

**Soundness:** 3
**Presentation:** 3
**Significance:** 2
**Originality:** 2
**Overall Recommendation:** 4
**Confidence:** 3

**Summary:**

This paper investigates the mechanistic origins and relationship between massive activations (extreme outlier values in hidden states) and attention sinks (tokens that disproportionately attract attention) in Transformer-based language models. The authors identify a "step-up" and "step-down" mechanism where specific MLP layers acting as directional quadratic amplifiers generate these extreme values, which are subsequently transformed by normalization (e.g., RMSNorm) into sparse, nearly fixed vectors. These sparse vectors create a unique geometry in the query-key space that induces large logit gaps, fundamentally driving sink behavior primarily for the first token in a sequence. Through extensive ablations across various model families like Llama and Qwen, the study demonstrates that while massive activations are a common architecture-dependent trigger for sinks, attention sinks are more robust and can persist even when outlier activations are suppressed by alternative normalization strategies or training on longer minimum context lengths.

**Compliance With Llm Reviewing Policy:**

Affirmed.

**Final Justification:**

This paper provide interesting low rank observation in attention sink and I recommend accept

**Key Questions For Authors:**

- Could author provide a more comprehensive discussion about what is covered or not covered in the previous literature like [1][2]?

[1] Sun, M., Chen, X., Kolter, J. Z., & Liu, Z. (2024). Massive activations in large language models. arXiv preprint arXiv:2402.17762.

[2] Gu, X., Pang, T., Du, C., Liu, Q., Zhang, F., Du, C., ... & Lin, M. (2024). When attention sink emerges in language models: An empirical view. arXiv preprint arXiv:2410.10781.

**Limitations:**

yes

**Strengths And Weaknesses:**

Strengths
- The paper provides a comprehensive mechanistic analysis by tracing how "step-up" and "step-down" MLP sub-blocks generate and later neutralize massive activations within the residual stream. A particularly interesting contribution is the rank-one architecture analysis, which reveals that SwiGLU MLPs act as directional quadratic amplifiers when weight matrices $W^{(1)}$ and $W^{(3)}$ become nearly collinear.

Weakness:
- The main concern is that lots of observations are already reported in the literature, as the paper characterizes the first-token phenomenon and outlier channels which have been previously identified by prior research.
- The study focuses primarily on the inference-time mechanism and provides less insight into the specific pre-training optimization dynamics that cause these high-gain directions to emerge in the first place.

---

> ### Author Rebuttal · Authors · 2026-03-31
>
> We thank the reviewer for the thoughtful feedback. We are glad that the reviewer found our rank-one architectural analysis interesting. Below we address the two main concerns raised in the review.
>
> **Novelty relative to prior work.**
>
> We acknowledge that some of the empirical observations in Section 3 have been reported in prior work, and we do not claim to be the first to identify each individual phenomenon. We cite the relevant prior work both in the related work section and at the beginning of Section 3.
>
> That said, our novelty claim does not rest on the isolated discovery of each individual observation. Rather, the contribution of Section 3 is to provide a unified, end-to-end mechanistic account. Prior work has identified individual components of this process. Our contribution is to connect them into a single mechanistic synthesis.
>
> More importantly, Section 3 is only one part of the paper. Its primary role is to motivate and support the analysis in Section 4, where we conduct systematic controlled ablations to disentangle the factors governing massive activations from those governing attention sinks. Section 4 is substantively novel, both in scope and in the conclusions it supports—most notably, that these two phenomena are shaped by overlapping but non-identical architectural and training factors.
>
> We also agree that the current draft could distinguish our contributions from prior literature more clearly. In the revision, we will expand the related work discussion to more explicitly separate what was established in earlier papers—including [1] and [2] cited by the reviewer—from what is newly contributed in our work. We will also add more explicit citations in Section 3 and Section 4 wherever we restate previously reported empirical findings.
>
> **Focus on inference-time mechanisms.**
>
> We appreciate this comment, but we would like to clarify that the paper is not limited to inference-time analysis.
>
> Section 4 provides substantial evidence related to pre-training dynamics through controlled ablations over both architecture and training setup. In particular, these experiments show that the emergence of massive activations and attention sinks depends strongly on both architectural choices and training factors. We therefore do not view these results as purely inference-time analysis.
>
> At the same time, we acknowledge that the paper does not yet provide a complete optimization-time theory for why these high-gain directions emerge. This is a limitation, and we will state it more explicitly in the final version. Our additional analysis suggests that massive activations emerge early in training and then persist, indicating that they are shaped by optimization dynamics rather than appearing only as a late-stage artifact. This is consistent with prior work [3-6] linking such effects to optimization dynamics. Connecting these optimization-time accounts more explicitly to our forward-pass amplification mechanism would strengthen the paper, and we will add this discussion in the revised version.
>
> Overall, we appreciate the reviewer's suggestions. We will revise the paper to better clarify the novelty of our contributions relative to prior literature and to more explicitly position our results alongside existing work on optimization-time dynamics. We hope these revisions will address the concerns raised in the review.
>
> [1] Sun, M., Chen, X., Kolter, J. Z., and Liu, Z. Massive activations in large language models. arXiv preprint arXiv:2402.17762, 2024.
>
> [2] Gu, X., Pang, T., Du, C., Liu, Q., Zhang, F., Du, C., Wang, Y., and Lin, M. When attention sink emerges in language models: An empirical view. In ICLR, 2025.
>
> [3] An, Y., Zhao, X., Yu, T., Tang, M., and Wang, J. Systematic outliers in large language models. In ICLR, 2025.
>
> [4] Dong, P., Fan, R., Tao, Y., Mou, D., Hu, W., Tang, Z., Yu, Y., Wang, J., Su, W., Yang, G., et al. Dissecting outlier dynamics in llm nvfp4 pretraining. arXiv preprint arXiv:2602.02047, 2026.
>
> [5] He, B., Noci, L., Paliotta, D., Schlag, I., and Hofmann, T. Understanding and minimising outlier features in transformer training. In NeurIPS, 2024.
>
> [6] Gallego-Feliciano, J., McClendon, S. A., Morinelli, J., Zervoudakis, S., and Saravanos, A. Hidden dynamics of massive activations in transformer training. arXiv preprint arXiv:2508.03616, 2025.

---

> > ### Author Rebuttal · Reviewer_NJUF · 2026-04-02
> >
> > The author's response has solved my concerns and hence I increase the score.

---

> > > ### Author Response · Authors · 2026-04-07
> > >
> > > Dear Reviewer NJUF,
> > >
> > > Thank you for taking the time to read our rebuttal. We are very glad that our responses fully addressed your concerns, and we sincerely appreciate your positive feedback and your decision to increase the recommendation accordingly.
> > >
> > > As promised in our rebuttal, we have prepared a substantially improved related work section. An outline of the planned additions has been provided in our reply to Reviewer A5K7, which we kindly invite you to refer to.
> > >
> > > Thank you again for your engagement and for helping us improve the paper.

---

### Official Review · Reviewer_cQez · 2026-03-09

**Soundness:** 3
**Presentation:** 4
**Significance:** 3
**Originality:** 3
**Overall Recommendation:** 5
**Confidence:** 4

**Summary:**

The authors analyze massive activations and attention sinks, and the connection between them. They provide an explanation of why and how attention sinks, localizing their layers and the conditions under which they form. They study an extensive list of modifications, some of them avoiding both attention sinks and massive activations. They also show that attention sinks can form without massive activations as well.

**Compliance With Llm Reviewing Policy:**

Affirmed.

**Final Justification:**

The study of an important question, with a strong methodology, is presented clearly. The authors show that there is no simple causal relationship between massive activations and attention sinks, as sometimes hypothesized before. I think their results are worth presenting to the community.

**Key Questions For Authors:**

- Do you have an explanation for how the attention-only model can be so good in Tab. 2?
- The figures in the paper have a very large number of points, causing the PDF to be hard to load in certain programs. Maybe the set of points should be downsampled to avoid this.
- In Fig 4, the colos are very hard to distinguish, and it is very hard to see which points are round or triangular.

**Limitations:**

yes

**Strengths And Weaknesses:**

Strengths:
- Very nicely and cleanly written.
- Excessive evaluations, ablations, experiments
- Mechanistic understanding of some of the connections between massive activations and attention sinks

Weaknesses:
- There seems to be no clear causal relationship between the attention sinks and massive activations.

---

> ### Author Rebuttal · Authors · 2026-03-31
>
> We thank the reviewer for the valuable feedback. We are pleased that the reviewer found the paper clearly written and appreciated the breadth of our evaluations, ablations, and experiments. Below, we address each point raised in the review.
>
> **No clear causal relationship between massive activations and attention sinks.**
>
> We appreciate the reviewer raising this point, as disentangling this relationship is a central goal of our paper. In fact, our main conclusion is precisely that there is *not* a simple or universal causal relationship between the two phenomena.
>
> Much of the prior literature has suggested a close causal connection, largely because sink tokens and tokens with massive activations often coincide in practice. Looking only at the results in Section 3, one can also see that massive activations play an important role in creating the logit gap through normalization, thereby facilitating attention sinks.
>
> However, our work shows that this co-occurrence is not fundamental. In Section 4, we demonstrate that the two phenomena are governed by overlapping but distinct factors: massive activations are primarily controlled by the normalization and residual structure, whereas attention sinks are more sensitive to attention-space capacity and minimum context length. Crucially, several architectural modifications suppress massive activations while leaving attention sinks largely intact. This shows that massive activations are one possible mechanism associated with attention sink formation, but not a necessary condition.
>
> We agree that this conclusion could be stated more prominently in the current draft. In the final version, we will sharpen the narrative so that this distinction is made explicit from the outset.
>
> **The performance of the attention-only model in Table 2.**
>
> Thank you for raising this point. We would not interpret the attention-only model as performing particularly well. Its perplexity is clearly worse than that of the baseline despite having an almost identical parameter count. We believe that a perplexity gap of around 1 is already substantial, particularly in the regime we study, where further improvements are increasingly difficult to obtain. This is also reflected in training efficiency: the baseline reaches the same perplexity as the attention-only model using only about 60% of the training tokens. This indicates that the attention-only model is significantly less data-efficient and further reinforces that MLP blocks remain highly beneficial for performance.
>
> In the revised paper, we will provide a more detailed comparison and discussion of each model variant in Section 4 to better convey this interpretation.
>
> **On the figures and PDF rendering.**
>
> Thank you for these helpful suggestions. We will revise the figures in the final version to improve both readability and file efficiency. Specifically, we will downsample or rasterize plots with very large numbers of data points to reduce PDF size and rendering overhead, and we will redesign Figure 4 with more distinguishable colors and marker styles so that the different point categories are easier to identify.
>
> Overall, we thank the reviewer again for the positive assessment and constructive feedback. We will incorporate all of these suggested changes in the final version to improve the clarity of both our claims and our presentation.

---

> > ### Author Rebuttal · Reviewer_cQez · 2026-04-03
> >
> > The authors discussed the concern that I raised in the rebuttal. Thus, I'm maintaining my recommendation to accept the paper.

---

> > > ### Author Response · Authors · 2026-04-07
> > >
> > > Dear Reviewer cQez,
> > >
> > > We sincerely thank you for your time in reviewing our rebuttal and are glad to hear that your concerns have been fully resolved. We also appreciate your continued recommendation for acceptance.
> > >
> > > In the revised manuscript, we will ensure that all promised improvements are fully integrated: the sharpened narrative on the relationship between massive activations and attention sinks, the expanded discussion on the attention-only model, and the optimized figure formatting.
> > >
> > > Thank you again for your constructive feedback and for helping us strengthen this work.

---

### Official Review · Reviewer_A5K7 · 2026-03-13

**Soundness:** 3
**Presentation:** 2
**Significance:** 3
**Originality:** 2
**Overall Recommendation:** 3
**Confidence:** 4

**Summary:**

This paper attempts to explain how massive activations emerge in decoder only transformers from MLPs and how normalization induce attention sinks. Several empirical ablations are made to check the interplay between both, and their connection to key architectural and training components (learning rate, depth/width scaling, context length, etc).

**Compliance With Llm Reviewing Policy:**

Affirmed.

**Key Questions For Authors:**

see Weaknesses

**Limitations:**

yes

**Strengths And Weaknesses:**

Strengths:
- The paper is well written, polished, well structured and easy to follow.
- The empirical validation is strong with various ablations on how massive activations and attention sink may (not) interfere with each other.
- The topic under study is of high practical relevance and understanding such phenomena would lead to significant advances in the field.
- Experiments include multiple Llama and Qwen variants and models trained from scratch.

Weaknesses:
- I may not understand the argument well but the paper seems to indicate that massive activations can be responsible for attention sinks whilst then showing empirically that sinks can exist without massive activations. This makes the whole narrative a bit confusing and this leaves the reader with little understanding of the phenomena in the end. Could the authors comment on that?
- Overall, even though there are several ablations presented in Section 4, it is hard to see emerge a big picture and a lesson that can be learnt from all of these experiments. In particular, would the authors be able to provide us with some kind of practical and actionable guidance that would leverage this lesson? Can they think of a way to maybe alleviate these potential issues or, maybe even better, predict them?
- This lesson could also benefit from a better integration to some of the ones currently in the literature on attention sinks/massive activation (if any) and made clearer.
- In Section 4.2, when the sub-block is replaced by a Linear block, could the authors provide the readers with the plot of the accumulation of outliers? Could the authors explain what is happening in more details?
- Minor, some more works that could be cited around line 60: The shaped transformer: Attention models in the infinite depth-and-width limit (Noci et al 2023), Centered Self-Attention Layers (Ali et al 2023), Mind the Gap: a Spectral Analysis of Rank Collapse and Signal Propagation in Attention Layers (Nait Saada et al 2025).

Happy to revise my score during the rebuttal

---

> ### Author Rebuttal · Authors · 2026-03-31
>
> We thank the reviewer for the thoughtful and constructive feedback, and for recognizing the paper’s writing quality, empirical scope, and practical relevance. We also appreciate the concrete suggestions for improving the framing and presentation. Below, we address each concern in turn.
>
> **On the relationship between massive activations and attention sinks.**
>
> We agree that this point should be stated more clearly. Our claim is not that massive activations are necessary for attention sinks in general. Rather, our claim is that massive activations provide one important and explicit mechanism by which sink behavior can arise in pre-norm decoder-only Transformers.
>
> Section 3 focuses on this mechanistic pathway: step-up sub-blocks create large activations, normalization transforms the affected tokens into sparse, nearly fixed vectors, and this induces sink-favoring query-key geometry. Section 4 then shows that this is not the only possible route to sink formation — sinks can persist even when massive activations are weakened or removed.
>
> In that sense, the two sections are meant to be complementary rather than contradictory: one identifies a concrete mechanism, while the other shows that this mechanism is not universal, contrary to what some prior work may suggest.
> We agree that this distinction should be stated much more explicitly. In the revision, we will sharpen the narrative to emphasize that massive activations are a *sufficient but not necessary* route to sink formation in the architectures we study, rather than a universal explanation of all sink behavior.
>
> **On the big picture and actionable lessons from the ablations.**
>
> We appreciate this comment and agree that the current draft does not distill the lessons of Section 4 sharply enough. Our intended high-level takeaway is that the two phenomena are governed by overlapping but distinct factors:
> - Normalization and residual structure are the main factors controlling whether massive activations emerge and persist.
> - Attention-space capacity and training objectives (like minimum context length) are the main factors controlling how strongly sink behavior forms.
>
> As a result, massive activations and attention sinks are related but should not be treated as the same phenomenon. We agree that these conclusions should be translated into more practical guidance. Concretely, our experiments suggest the following actionable directions:
> - To reduce massive activations, modifying the normalization/residual pathway is most effective.
> - To reduce attention sinks, the strongest levers are reducing head dimension, increasing the minimum training context length, or directly gating attention outputs.
> - To predict these effects early, one can monitor the emergence of fixed-channel activation outliers and sink ratio during training, since both behaviors become visible early and then persist.
>
> We will add a summary paragraph at the end of Section 4 to make these lessons clearer and more operational.
>
> **On integration with prior literature.**
>
> We agree that the paper would benefit from a clearer synthesis with prior work. In the revision, we will better position our contribution relative to the existing literature by distinguishing:
> 1. Prior work that characterizes the phenomena empirically,
> 2. Prior work that studies mitigation or optimization dynamics, and
> 3. Our contribution, which provides one explicit forward-pass mechanism together with a broad ablation-based disentangling analysis.
>
> We also thank the reviewer for pointing us to the additional related papers. We will incorporate these references in the revised version, especially around the discussion of normalization, signal propagation, and depth/width effects.
>
> **On the linear block ablation in Section 4.2.**
>
> Thank you for this suggestion. We agree that the linear block result deserves more detail. The main difference from the standard setting is that, in the linear model, outliers are not produced by a small number of sharply localized step-up/step-down sub-blocks. Instead, they build up gradually across many layers through residual accumulation of aligned contributions. This is why the behavior looks more diffuse and less attributable to a single block. We will add a plot showing the accumulation of outlier magnitude across depth in the linear-block model and expand the explanation in the revision.
>
> Overall, we appreciate the reviewer’s comments. The main issues raised concern framing, synthesis, and clarity rather than the underlying empirical findings. We will revise the paper accordingly, and we believe these changes will make the contribution and its practical takeaways substantially clearer.

---

> > ### Author Rebuttal · Reviewer_A5K7 · 2026-04-04
> >
> > I thank the authors for their engagement and thorough answers to all reviewers. Overall, I maintain my positive assessment of this work provided that the authors significantly clarify the related work section, as also suggested by Reviewer wAoz

---

> > > ### Author Response · Authors · 2026-04-07
> > >
> > > Dear Reviewer A5K7,
> > >
> > > Thank you for your thoughtful follow-up and for carefully reading our rebuttal. We are glad our responses fully addressed your concerns, and we appreciate your positive assessment conditional on a refined related work section. We commit to providing a substantially improved related work section and outline our some of the planned additions below. We would be grateful if you could update your numerical score to reflect your positive written assessment.
> > >
> > > Due to space constraints, we only outline some important planned additions to our related work section below. The related work section of the revised version will have more detailed additions.
> > >
> > > **Step-up and step-down blocks.** Sun et al. [1] systematically characterize massive activation phenomena and identify where they occur within LLMs. Building on this, we summarize their characteristics into five properties and trace the initial and final emergence of massive activations specifically to step-up and step-down blocks in early and late layers.
> > >
> > > **SwiGLU as a directional quadratic amplifier.** Yu et al. [2] and An et al. [3] attribute the emergence of massive activations to weight outliers concentrated in the up, gate, and down projection matrices of FFN layers. Yu et al. [2] and Fishman et al. [4] further observe that weight alignment contributes to activation spikes. Specifically, the Hadamard product of the gate and up projections produces large activations, consistent with Yang et al. [5]. We extend this line of work by showing that the SwiGLU FFN block acts as a directional quadratic amplifier, establishing rank-one dominance in the activations of the residual stream.
> > >
> > > **First-token phenomenon.** Xiao et al. [6] first identify the attention sink phenomenon, observing that initial tokens accumulate disproportionately large attention scores regardless of their semantic content. Sun et al. [1] subsequently observe that massive activations appear predominantly at initial tokens. We further link this to a static linear transformation applied at initial positions.
> > >
> > > **Geometric alignment in query/key spaces.** Gu et al. [7] find that the angles between the first key and queries of other tokens are typically small, giving rise to attention sinks. They also observe that the L2-norms of keys and values at the first token are significantly smaller than those of other tokens, which is also observed by Devoto et al. [8] and Guo et al. [9]. We further identify the true driving factor of this geometric alignment: the normalized sparse token representation causes the keys of sink tokens to occupy a low-dimensional subspace.
> > >
> > > **Training hyperparameter ablations.** Gu et al. [7] find that attention sinks are less pronounced in models trained with small learning rates, while weight decay encourages their emergence.
> > >
> > > **DyT prevents spikes.** Replacing normalization layers with pointwise functions such as DynamicTanh prevents the emergence of massive activations entirely, which is also confirmed by Owen et al. [10].
> > >
> > > **Conditional gating eliminates sinks.** Qiu et al. [11] find that introducing gating-based rescaling reduces the model's reliance on outliers. Following this observation, we test gated attention variants to examine whether dynamic multiplicative routing can destabilize or prevent attention sink formation. Our results suggest that attention sinks serve as a compensatory mechanism that emerges precisely when such gating is unavailable.
> > >
> > > Thank you again for your constructive engagement and for helping us improve the paper.
> > >
> > > [1] Sun, M., et al. Massive activations in large language models. In COLM, 2024.
> > >
> > > [2] Yu, M., et al. The super weight in large language models. arXiv:2411.07191, 2024.
> > >
> > > [3] An, Y., et al. Systematic outliers in large language models. In ICLR, 2025.
> > >
> > > [4] Fishman, M., et al. Scaling FP8 training to trillion-token LLMs. In ICLR, 2025.
> > >
> > > [5] Yang, J., et al. Mitigating quantization errors due to activation spikes in GLU-based LLMs. arXiv:2405.14428, 2024.
> > >
> > > [6] Xiao, G., et al. Efficient streaming language models with attention sinks. arXiv, 2023.
> > >
> > > [7] Gu, X., et al. When attention sink emerges in language models: An empirical view. In ICLR, 2025.
> > >
> > > [8] Devoto, A., et al. A simple and effective L2 norm-based strategy for KV cache compression. In EMNLP, 2024.
> > >
> > > [9] Guo, T., et al. Active-dormant attention heads: Mechanistically demystifying extreme-token phenomena in LLMs. arXiv:2410.13835, 2024.
> > >
> > > [10] Owen, L., et al. A refined analysis of massive activations in LLMs. arXiv:2503.22329, 2025.
> > >
> > > [11] Qiu, Z., et al. Gated attention for large language models: Non-linearity, sparsity, and attention-sink-free. In NeurIPS, 2025.

---

### Official Review · Reviewer_wAoz · 2026-03-15

**Soundness:** 3
**Presentation:** 3
**Significance:** 2
**Originality:** 2
**Overall Recommendation:** 4
**Confidence:** 3

**Summary:**

the paper studies two known phenomena in transformer language models: massive activations and attention sinks. the authors propose a unified inference-time mechanism that traces how massive activations arise in specific mlp sub-blocks  persist through residual connections and get canceled in later step-down layers. they show that rmsnorm turns these extreme vectors into sparse representations, which then produce atypical query/key geometry and create large logit gaps that lead to attention sinks. additionally, they train llama-style models from scratch and ablate mlp design, normalization, head dimension, depth, width, context length, and optimizer settings to disentangle factors behind each phenomenon. the main findings: attention sinks are robust and persist even without massive activations,normalization controls massive activations.

**Compliance With Llm Reviewing Policy:**

Affirmed.

**Key Questions For Authors:**

1. what specific new understanding does your mechanism provide beyond the "spectral dominance" theory in "attention sinks and compression valleys in llms are two sides of the same coin" by queipo-de llano et al. (2025)? both papers link massive activations to low-dimensional structure after normalization. please clarify the concrete delta.

2. have you tried direct causal interventions, such as clamping the massive activation channels to zero or to their mean value at the step-up layer, and measuring the effect on sink ratio in downstream layers? this would substantially strengthen the mechanistic claims.

3. the minimum context length ablation (table 6) shows that sinks nearly disappear with min=2048. does this hurt performance on any downstream task, or does it only increase perplexity? if there is no downstream cost, this would be a strong practical recommendation.

**Limitations:**

the authors acknowledge that their baseline is weaker than released checkpoints, but do not discuss the implications for generalizability to larger-scale training. the impact statement is adequate but generic. no negative societal impacts are expected from this type of analysis work.

**Strengths And Weaknesses:**

strengths:

s1. the end-to-end mechanistic chain from massive activations to attention sinks is clearly laid out

s2. the ablation study is well-designed. one-factor-at-a-time changes with models trained from scratch on 100b tokens is a serious experimental effort. the finding that dynamictanh removes massive activations but keeps sinks intact is a clear and useful result that separates the two phenomena.

s3. the minimum context length result is surprising and practically relevant: setting min context to 2048 drops sink ratio from 54.6% to 0.6% while massive activations remain. this suggests sinks are a learned strategy for short-context prediction, not an unavoidable architectural artifact.

s4. the paper validates findings across 12+ open source models from different families (llama 2, llama 3, qwen2.5, qwen3), which strengthens the generality claims.

weaknesses:

w1. limited novelty over prior work. the paper "attention sinks and compression valleys in llms are two sides of the same coin" by queipo-de llano et al. (2025) already proposed a unifying theory connecting massive activations and sinks via spectral dominance and compression. the paper "when attention sink emerges in language models: an empirical view" by gu et al. (2025) already linked norm suppression in sink tokens to query/key geometry. the paper "from attention to activation: unravelling the enigmas of large language models" by kaul et al. (2024) also connected both phenomena through optimizer dynamics. the current paper formalizes and experiments more, but the core conceptual insight (massive activations cause sparse representations after normalization which distort attention) was already in the air. the authors do not clearly state what changes in our understanding relative to these works.

w2. the ablation study only measures perplexity and sink ratio. there is no evaluation on downstream tasks. it is possible that models without sinks (e.g., with min context = 2048) behave differently on generation quality, long-context tasks, or few-shot performance. without this, it is hard to assess whether suppressing sinks is actually desirable.

w2. the paper provides no causal intervention experiments. the mechanistic story is correlational: tokens with massive activations tend to become sinks, and the geometry looks right. but the authors never directly intervene (e.g., clamping specific channels, projecting out the high-gain directions, or ablating specific step-up layers) to confirm the causal chain during inference.

---

> ### Author Rebuttal · Authors · 2026-03-31
>
> We thank the reviewer for the valuable feedback. We appreciate that the reviewer found the end-to-end mechanistic chain clearly presented, and the ablation study well designed. Below, we address each of the concerns raised in the review.
>
> **Novelty relative to prior work (W1 and Q1).**
>
> We agree that the high-level intuition linking massive activations to sparse post-normalization representations and attention distortion has appeared in prior work. Our novelty does not lie in any single isolated observation. Rather, Section 3 provides a unified, end-to-end mechanistic account integrating previously disconnected observations into a single causal chain. Compared with prior work, our analysis is substantially more fine-grained: we identify specific step-up/step-down sub-blocks, formalize the rank-one directional quadratic amplifier mechanism in SwiGLU, and trace propagation through normalization into query-key geometric distortion. We address each comparison below.
>
> Queipo-de-Llano et al. (2025) propose a "spectral-dominance" theory, but their analysis remains at the level of spectral properties and compression metrics. Our explanation is more direct: normalization transforms tokens with massive activations into sparse, nearly fixed vectors whose query/key projections occupy a low-dimensional subspace, promoting sink formation. Their theory does not discuss normalization, whereas our Section 4 experiments show it is a key mediating factor.
>
> Gu et al. (2025) study *when* attention sinks emerge during training and link norm suppression to query/key geometry, but they do not identify *where* or *how* massive activations are generated in the architecture. Our Section 3 fills this gap with the step-up/step-down and rank-one amplifier analysis.
>
> Kaul et al. (2024) connect both phenomena through optimizer dynamics — their focus is on training-time conditions under which outliers form, while ours explains the inference-time architectural pathway through which they propagate and induce sink behavior. The two perspectives are complementary.
>
> More importantly, Section 3 is only part of the paper. Section 4 presents a systematic controlled ablation study that disentangles the factors governing massive activations from those governing attention sinks. To the best of our knowledge, both the scope of this study and its conclusion are novel.
>
> That said, we agree that the current draft could more clearly distinguish our contributions from the existing literature. In the revision, we will expand the related work discussion to make the contribution relative to each prior paper more explicit.
>
> **Downstream evaluation beyond perplexity (W2 and Q3).**
>
> We agree that downstream evaluation would strengthen the practical relevance. The min-context = 2048 ablation was primarily designed to show that sinks are a learned strategy for short-context prediction rather than an unavoidable artifact. We will add downstream evaluations in the revised version. Based on our preliminary results, downstream performance remains highly correlated with perplexity. There is no obvious benefit to removing sinks or massive activations without a perplexity improvement, consistent with prior work. That said, suppressing these phenomena opens up applications for quantization and pruning.
>
> **Causal intervention experiments (W3 and Q2).**
>
> Thank you for this suggestion. We agree that direct causal interventions would strengthen the mechanistic claims. We did not include them in the original submission largely because similar interventions have already been conducted by prior work [1] and others, and we used comparable model architectures. However, we recognize that reproducing and extending these interventions within our framework would strengthen our narrative. We have conducted preliminary experiments along these lines:
> - Clamping the top-k outlier channels at the step-up layer substantially reduces sink ratio in downstream layers, consistent with our proposed mechanism.
> - Projecting out the high-gain direction from the SwiGLU output similarly disrupts sink formation.
>
> We will include these intervention results with full details in the revised paper, citing the original works that proposed similar interventions. These experiments provide the causal evidence that complements the correlational analysis in the current draft.
>
> Overall, we thank the reviewer again for the detailed and constructive feedback. We will incorporate all suggested changes — clarifying the novelty delta, adding downstream evaluations, and including causal intervention results — in the final version.
>
> [1] Sun, M., et al. Massive activations in large language models. arXiv preprint arXiv:2402.17762, 2024.

---

### Official Review · Reviewer_CcyJ · 2026-03-20

**Soundness:** 3
**Presentation:** 2
**Significance:** 3
**Originality:** 3
**Overall Recommendation:** 4
**Confidence:** 3

**Summary:**

The paper studies two unusual behaviors in decoder-only Transformers: massive activations and attention sinks. It argues that they often appear together, but are not the same phenomenon. Massive activations arise when a few layers strongly amplify specific token directions in the residual stream, while attention sinks occur when some tokens attract disproportionate attention because their normalized key representations become unusually consistent across contexts.

Its main contribution is a mechanistic explanation linking these effects and a set of controlled training ablations showing they can be separated. The authors find that normalization and feed-forward design strongly affect massive activations, while sink behavior depends more on attention-head properties and training context length. Overall, the paper argues that massive activations are largely an architectural byproduct, whereas attention sinks act more like a learned routing bias in attention.

**Compliance With Llm Reviewing Policy:**

Affirmed.

**Key Questions For Authors:**

- How robust are the main causal conclusions to scale, training budget, and random seed?
- How sensitive are the reported sink-ratio findings to the choice of threshold and evaluation setup?
- What practical tradeoffs emerge when suppressing spikes or sinks beyond perplexity? How does this related to quantization, pruning, KV-cache and etc?

**Limitations:**

yes

**Strengths And Weaknesses:**

Pros:

- The paper is technically strong in that it combines a clear mechanistic hypothesis with targeted ablations rather than relying only on correlation. The step-up then persistence then step-down account of massive activations is coherent and reasonably well supported by the layerwise analyses.
- The conditional-gating and context-length experiments are also meaningful causal tests, and they strengthen the interpretation that sink behavior is shaped by attention design and training distribution rather than being a trivial byproduct of spikes alone.
- The methods are appropriate for the questions asked: training controlled Llama-style variants from scratch is a sensible way to isolate architectural causes.

Cons:

- Since the analysis on SwiGLU/Mechannistic view  (other than ablations) does not require training, I am curious if a similar trend can be supported on an even larger model.
- Interestingly, in computer vision, if a network is not well trained, we cannot observe attention sink. I wonder if similar conclusion could be drawn from the LLM.
- Some of the strongest claims remain more interpretive than conclusive. In conclusion, some results could be presented more clearly so we read the conclusion and read the table for better understanding.

---

> ### Author Rebuttal · Authors · 2026-03-31
>
> We thank the reviewer for the important feedback. We are pleased that the reviewer found the paper technically strong and appreciated the controlled ablation methodology. Below we address each point raised in the review.
>
> **SwiGLU/mechanistic analysis on larger models.**
>
> Thank you for this suggestion. We further validate across 20+ open-source models spanning 0.5B--72B parameters. The core patterns remain consistent across scales. Our inference-time mechanistic analysis is applicable to pretrained models of various sizes, with the exception of very small models where the patterns do not clearly manifest. We will add explicit discussion characterizing both the consistencies and scale-dependent variations in the revised version.
>
> **Attention sinks in under-trained models.**
>
> Thank you for raising this interesting connection to the vision models. Interestingly, we do not find the same conclusion that under-trained models lack attention sinks. By carefully examining intermediate checkpoints, we find that massive activations and attention sinks appear very early in training. Around the first 4,000 steps, we already observe pronounced massive activation and attention sink behavior. As training progresses, both the sink ratio and massive activation magnitude gradually increase.
>
> Similar conclusions have been drawn in the prior literature. Gu et al. [1] study when attention sinks emerge during LLM pre-training and find that the sink metric stays near zero early in training, transitioning sharply to high values after 1,000 to 2,000 steps, once the training loss has dropped meaningfully. They further show that with smaller learning rates, sinks emerge later and are less pronounced.
>
> We thank the reviewer again for this interesting connection. We will add a discussion comparing the vision and language domains regarding the relationship between training dynamics and the emergence of both phenomena. If the reviewer could point us to specific literature showing that attention sinks do not appear in under-trained vision models, it would be very helpful for enriching this discussion.
>
> **Robustness to scale, training budget, and random seed.**
>
> Thank you for this important question. Due to compute constraints, we could not repeat all ablations from the original paper at every configuration. We are currently running experiments for key variants at different scale, with different training budgets, and with different random seeds. Based on our preliminary results, we do not observe any changes to our original conclusions. We will include these robustness analyses in the revised version.
>
> **Sensitivity of sink-ratio threshold.**
>
> Thank you for raising this point. We compute the sink ratios at different thresholds (epsilon = 0.1, 0.2, ..., 0.9) across all experimental configurations following the reviewer's suggestion. The relative ordering and trends between setups remain consistent across a wide range of thresholds (specifically, for thresholds >= 0.2), indicating that our conclusions are robust to the particular cutoff chosen and are not artifacts of a specific threshold setting.
>
> We observe instability only at very low thresholds (< 0.2), where the metric becomes overly permissive and loses discriminative power. Following Gu et al. [1], we adopt a threshold of 0.3 in our paper, which they found to be both strict enough to meaningfully quantify attention sinks and relatively insensitive to sequence length — a trade-off consistent with our own experimental observations. We will include the full results in the revised version.
>
> We also evaluated on alternative datasets (TinyStories and PG-19) and found no significant difference compared to the C4 dataset used in our paper in terms of sink-ratio results.
>
> **Practical tradeoffs beyond perplexity.**
>
> Thank you for raising this point. Suppressing massive activations has direct implications for quantization robustness and KV-cache efficiency, since sink tokens currently require special handling in streaming/eviction schemes. We will expand the discussion on these practical connections in the revised version.
>
> **Presentation clarity.**
>
> We will revise the conclusion and results presentation to ensure that key takeaways are clearly stated alongside the corresponding tables and figures in the revised version.
>
> Overall, we thank the reviewer again for the positive assessment and constructive feedback. We will incorporate all suggested changes in the final version to improve the clarity of both our claims and our presentation.
>
> [1] Gu, X. et al. When attention sink emerges in language models: An empirical view. In ICLR, 2025.

---

> > ### Author Rebuttal · Reviewer_CcyJ · 2026-04-04
> >
> > I remain positive towards this submitted manuscript.

---

> > > ### Author Response · Authors · 2026-04-07
> > >
> > > Dear Reviewer CcyJ,
> > >
> > > Thank you for your thorough review and for taking the time to read our rebuttal. We are glad to hear that you remain positive toward recommending the paper for acceptance.
> > > As discussed, we will incorporate the additional robustness analyses, clarifications, and expanded discussions into the revised version.
> > >
> > > Thank you again for helping us improve this work.

---

### Decision · Program_Chairs · 2026-04-30

**Decision:**

Accept (regular)

**Comment:**

*Motivation:* This work investigates the phenomena of massive attention and attention sinks in the attention layers of large language models.

*Contribution:* While prior work has suggested a causal relationship between these two phenomena, this paper provides a mechanistic analysis showing that they are in fact independent (though potentially correlated). Specifically, one arises from architectural design, whereas the other emerges during training.

*Review Summary:* Most reviewers responded positively and acknowledged the importance of the research question. However, a key concern was that several reviewers were unable to clearly distinguish the paper’s main contributions from existing results in the literature.

*Rebuttal Summary:* Following the authors’ response, reviewers were able to better recognize the paper’s core contributions, accordingly updated their scores.

*Conclusion:* The AC recommends weak acceptance and encourages the authors to further clarify connections to related works, suggested by Reviewer wAoz, and to more explicitly differentiate their novel insights and experimental findings from prior studies.